# Domain Adaptation with Conditional Distribution Matching and Generalized Label Shift

**Remi Tachet des Combes**[*]
Microsoft Research Montreal
Montreal, QC, Canada
retachet@microsoft.com

**Han Zhao**[*]
D. E. Shaw & Co.
New York, NY, USA
han.zhao@cs.cmu.edu

**Yu-Xiang Wang**
UC Santa Barbara
Santa Barbara, CA, USA
yuxiangw@cs.ucsb.edu

**Geoff Gordon**
Microsoft Research Montreal
Montreal, QC, Canada
ggordon@microsoft.com

## Abstract

Adversarial learning has demonstrated good performance in the unsupervised domain adaptation setting, by learning domain-invariant representations. However, recent work has shown limitations of this approach when label distributions differ between the source and target domains. In this paper, we propose a new assumption, *generalized label shift* (*GLS*), to improve robustness against mismatched label distributions. *GLS* states that, conditioned on the label, there exists a representation of the input that is invariant between the source and target domains. Under *GLS*, we provide theoretical guarantees on the transfer performance of any classifier. We also devise necessary and sufficient conditions for *GLS* to hold, by using an estimation of the relative class weights between domains and an appropriate reweighting of samples. Our weight estimation method could be straightforwardly and generically applied in existing domain adaptation (DA) algorithms that learn domain-invariant representations, with small computational overhead. In particular, we modify three DA algorithms, JAN, DANN and CDAN, and evaluate their performance on standard and artificial DA tasks. Our algorithms outperform the base versions, with vast improvements for large label distribution mismatches. Our code is available at https://tinyurl.com/y585xt6j.

## 1   Introduction

In spite of impressive successes, most deep learning models [22] rely on huge amounts of labelled data and their features have proven brittle to distribution shifts [39, 55]. Building more robust models, that learn from fewer samples and/or generalize better out-of-distribution is the focus of many recent works [2, 5, 53]. The research direction of interest to this paper is that of domain adaptation, which aims at learning features that transfer well between domains. We focus in particular on unsupervised domain adaptation (UDA), where the algorithm has access to labelled samples from a source domain and unlabelled data from a target domain. Its objective is to train a model that generalizes well to the target domain. Building on advances in adversarial learning [23], adversarial domain adaptation (ADA) leverages the use of a discriminator to learn an intermediate representation that is invariant between the source and target domains. Simultaneously, the representation is paired with a classifier, trained to perform well on the source domain [20, 32, 49, 60]. ADA is rather successful on a variety

---

[*]The first two authors contributed equally to this work.  Work done while HZ was at Carnegie Mellon University.

of tasks, however, recent work has proven an upper bound on the performance of existing algorithms when source and target domains have mismatched label distributions [62]. Label shift is a property of two domains for which the marginal label distributions differ, but the conditional distributions of input given label stay the same across domains [48, 58].

In this paper, we study domain adaptation under mismatched label distributions and design methods that are robust in that setting. Our contributions are the following. First, we extend the upper bound by Zhao et al. [62] to $k$-class classification and to conditional domain adversarial networks, a recently introduced domain adaptation algorithm [37]. Second, we introduce *generalized label shift* (*GLS*), a broader version of the standard label shift where conditional invariance between source and target domains is placed in representation rather than input space. Third, we derive performance guarantees for algorithms that seek to enforce *GLS* via learnt feature transformations, in the form of upper bounds on the error gap and the joint error of the classifier on the source and target domains. Those guarantees suggest principled modifications to ADA to improve its robustness to mismatched label distributions. The modifications rely on estimating the class ratios between source and target domains and use those as importance weights in the adversarial and classification objectives. The importance weights estimation is performed using a method of moment by solving a quadratic program, inspired from Lipton et al. [31]. Following these theoretical insights, we devise three new algorithms based on learning importance-weighted representations, DANNs [20], JANs [36] and CDANs [37]. We apply our variants to artificial UDA tasks with large divergences between label distributions, and demonstrate significant performance gains compared to the algorithms' base versions. Finally, we evaluate them on standard domain adaptation tasks and also show improved performance.

## 2 Preliminaries

**Notation** We focus on the general $k$-class classification problem. $\mathcal{X}$ and $\mathcal{Y}$ denote the input and output space, respectively. $\mathcal{Z}$ stands for the representation space induced from $\mathcal{X}$ by a feature transformation $g : \mathcal{X} \mapsto \mathcal{Z}$. Accordingly, we use $X, Y, Z$ to denote random variables which take values in $\mathcal{X}, \mathcal{Y}, \mathcal{Z}$. *Domain* corresponds to a joint distribution on the input space $\mathcal{X}$ and output space $\mathcal{Y}$, and we use $\mathcal{D}_S$ (resp. $\mathcal{D}_T$) to denote the source (resp. target) domain. Noticeably, this corresponds to a stochastic setting, which is stronger than the deterministic one previously studied [6, 7, 62]. A *hypothesis* is a function $h : \mathcal{X} \to [k]$. The *error* of a hypothesis $h$ under distribution $\mathcal{D}_S$ is defined as: $\varepsilon_S(h) := \Pr_{\mathcal{D}_S}(h(X) \neq Y)$, i.e., the probability that $h$ disagrees with $Y$ under $\mathcal{D}_S$.

**Domain Adaptation via Invariant Representations** For source ($\mathcal{D}_S$) and target ($\mathcal{D}_T$) domains, we use $\mathcal{D}_S^X, \mathcal{D}_T^X, \mathcal{D}_S^Y$ and $\mathcal{D}_T^Y$ to denote the marginal data and label distributions. In UDA, the algorithm has access to $n$ labeled points $\{(\mathbf{x}_i, y_i)\}_{i=1}^n \in (\mathcal{X} \times \mathcal{Y})^n$ and $m$ unlabeled points $\{\mathbf{x}_j\}_{j=1}^m \in \mathcal{X}^m$ sampled i.i.d. from the source and target domains. Inspired by Ben-David et al. [7], a common approach is to learn representations invariant to the domain shift. With $g : \mathcal{X} \mapsto \mathcal{Z}$ a feature transformation and $h : \mathcal{Z} \mapsto \mathcal{Y}$ a hypothesis on the feature space, a domain invariant representation [20, 49, 59] is a function $g$ that induces similar distributions on $\mathcal{D}_S$ and $\mathcal{D}_T$. $g$ is also required to preserve rich information about the target task so that $\varepsilon_S(h \circ g)$ is small. The above process results in the following Markov chain (assumed to hold throughout the paper):

$$X \xrightarrow{g} Z \xrightarrow{h} \widehat{Y}, \tag{1}$$

where $\widehat{Y} = h(g(X))$. We let $\mathcal{D}_S^Z, \mathcal{D}_T^Z, \mathcal{D}_S^{\widehat{Y}}$ and $\mathcal{D}_T^{\widehat{Y}}$ denote the pushforwards (induced distributions) of $\mathcal{D}_S^X$ and $\mathcal{D}_T^X$ by $g$ and $h \circ g$. Invariance in feature space is defined as minimizing a distance or divergence between $\mathcal{D}_S^Z$ and $\mathcal{D}_T^Z$.

**Adversarial Domain Adaptation** Invariance is often attained by training a discriminator $d : \mathcal{Z} \mapsto [0, 1]$ to predict if $z$ is from the source or target. $g$ is trained both to maximize the discriminator loss and minimize the classification loss of $h \circ g$ on the source domain ($h$ is also trained with the latter objective). This leads to domain-adversarial neural networks [20, DANN], where $g$, $h$ and $d$ are parameterized with neural networks: $g_\theta$, $h_\phi$ and $d_\psi$ (see Algo. 1 and App. **??**). Building on DANN, conditional domain adversarial networks [37, CDAN] use the same adversarial paradigm. However, the discriminator now takes as input the outer product, for a given $x$, between the predictions of the network $h(g(x))$ and its representation $g(x)$. In other words, $d$ acts on the outer product: $h \otimes g(x) := (h_1(g(x)) \cdot g(x), \ldots, h_k(g(x)) \cdot g(x))$ rather than on $g(x)$. $h_i$ denotes the $i$-th element of vector $h$. We now highlight a limitation of DANNs and CDANs.

Table 1: Common assumptions in the domain adaptation literature.

| Covariate Shift | Label Shift |
|---|---|
| $\mathcal{D}_S^X \neq \mathcal{D}_T^X$ | $\mathcal{D}_S^Y \neq \mathcal{D}_T^Y$ |
| $\forall \mathbf{x} \in \mathcal{X}, \mathcal{D}_S(Y \mid X = \mathbf{x}) = \mathcal{D}_T(Y \mid X = \mathbf{x})$ | $\forall y \in \mathcal{Y}, \mathcal{D}_S(X \mid Y = y) = \mathcal{D}_T(X \mid Y = y)$ |

**An Information-Theoretic Lower Bound** We let $D_{\mathrm{JS}}$ denote the Jensen-Shanon divergence between two distributions (App. **??**), and $\widetilde{Z}$ correspond to $Z$ (for DANN) or to $\widehat{Y} \otimes Z$ (for CDAN). The following theorem lower bounds the joint error of the classifier on the source and target domains:
**Theorem 2.1.** Assuming that $D_{\mathrm{JS}}(\mathcal{D}_S^Y \parallel \mathcal{D}_T^Y) \geq D_{\mathrm{JS}}(\mathcal{D}_S^{\widetilde{Z}} \parallel \mathcal{D}_T^{\widetilde{Z}})$, then:

$$\varepsilon_S(h \circ g) + \varepsilon_T(h \circ g) \geq \frac{1}{2}\left(\sqrt{D_{\mathrm{JS}}(\mathcal{D}_S^Y \parallel \mathcal{D}_T^Y)} - \sqrt{D_{\mathrm{JS}}(\mathcal{D}_S^{\widetilde{Z}} \parallel \mathcal{D}_T^{\widetilde{Z}})}\right)^2 .$$

**Remark** The lower bound is algorithm-independent. It is also a population-level result and holds asymptotically with increasing data. Zhao et al. [62] prove the theorem for $k = 2$ and $\widetilde{Z} = Z$. We extend it to CDAN and arbitrary $k$ (it actually holds for any $\widetilde{Z}$ s.t. $\widehat{Y} = \widetilde{h}(\widetilde{Z})$ for some $\widetilde{h}$, see App. **??**). Assuming that label distributions differ between source and target domains, the lower bound shows that: *the better the alignment of feature distributions, the worse the joint error.* For an invariant representation ($D_{\mathrm{JS}}(\mathcal{D}_S^{\widetilde{Z}}, \mathcal{D}_T^{\widetilde{Z}}) = 0$) with no source error, the target error will be larger than $D_{\mathrm{JS}}(\mathcal{D}_S^Y, \mathcal{D}_T^Y)/2$. Hence algorithms learning invariant representations and minimizing the source error are fundamentally flawed when label distributions differ between source and target domains.

**Common Assumptions to Tackle Domain Adaptation** Two common assumptions about the data made in DA are *covariate shift* and *label shift*. They correspond to different ways of decomposing the joint distribution over $X \times Y$, as detailed in Table 1. From a representation learning perspective, covariate shift is not robust to feature transformation and can lead to an effect called negative transfer [62]. At the same time, label shift clearly fails in most practical applications, e.g. transferring knowledge from synthetic to real images [51]. In that case, the input distributions are actually disjoint.

## 3 Main Results

In light of the limitations of existing assumptions, (e.g. covariate shift and label shift), we propose *generalized label shift* ($GLS$), a relaxation of label shift that substantially improves its applicability. We first discuss some of its properties and explain why the assumption is favorable in domain adaptation based on representation learning. Motivated by $GLS$, we then present a novel error decomposition theorem that directly suggests a bound minimization framework for domain adaptation. The framework is naturally compatible with $\mathcal{F}$-integral probability metrics [40, $\mathcal{F}$-IPM] and generates a family of domain adaptation algorithms by choosing various function classes $\mathcal{F}$. In a nutshell, the proposed framework applies a method of moments [31] to estimate the importance weight $\mathbf{w}$ of the *marginal label distributions* by solving a quadratic program (QP), and then uses $\mathbf{w}$ to align the weighted source feature distribution with the target feature distribution.

### 3.1 Generalized Label Shift

**Definition 3.1** (Generalized Label Shift, $GLS$)**.** A representation $Z = g(X)$ satisfies $GLS$ if

$$\mathcal{D}_S(Z \mid Y = y) = \mathcal{D}_T(Z \mid Y = y), \ \forall y \in \mathcal{Y}. \tag{2}$$

First, when $g$ is the identity map, the above definition of $GLS$ reduces to the original label shift assumption. Next, $GLS$ is always achievable for any distribution pair $(\mathcal{D}_S, \mathcal{D}_T)$: any constant function $g \equiv c \in \mathbb{R}$ satisfies the above definition. The most important property is arguably that, unlike label shift, $GLS$ is compatible with a perfect classifier (in the noiseless case). Precisely, if there exists a ground-truth labeling function $h^*$ such that $Y = h^*(X)$, then $h^*$ satisfies $GLS$. As a comparison, without conditioning on $Y = y$, $h^*$ does not satisfy $\mathcal{D}_S(h^*(X)) = \mathcal{D}_T(h^*(X))$ if the marginal label distributions are different across domains. This observation is consistent with the lower bound in Theorem 2.1, which holds for arbitrary marginal label distributions.

*GLS* imposes label shift in the feature space $\mathcal{Z}$ instead of the original input space $\mathcal{X}$. Conceptually, although samples from the same classes in the source and target domain can be dramatically different, the hope is to find an intermediate representation for both domains in which samples from a given class look similar to one another. Taking digit classification as an example and assuming the feature variable $Z$ corresponds to the contour of a digit, it is possible that by using different contour extractors for e.g. MNIST and USPS, those contours look roughly the same in both domains. Technically, *GLS* can be facilitated by having separate representation extractors $g_S$ and $g_T$ for source and target [9, 49].

## 3.2 An Error Decomposition Theorem based on *GLS*

We now provide performance guarantees for models that satisfy *GLS*, in the form of upper bounds on the error gap and on the joint error between source and target domains. It requires two concepts:

**Definition 3.2** (Balanced Error Rate). The *balanced error rate* (BER) of predictor $\widehat{Y}$ on $\mathcal{D}_S$ is:

$$\mathrm{BER}_{\mathcal{D}_S}(\widehat{Y} \parallel Y) := \max_{j \in [k]} \mathcal{D}_S(\widehat{Y} \neq Y \mid Y = j). \tag{3}$$

**Definition 3.3** (Conditional Error Gap). Given a joint distribution $\mathcal{D}$, the *conditional error gap* of a classifier $\widehat{Y}$ is $\Delta_{\mathrm{CE}}(\widehat{Y}) := \max_{y \neq y' \in \mathcal{Y}^2} |\mathcal{D}_S(\widehat{Y} = y' \mid Y = y) - \mathcal{D}_T(\widehat{Y} = y' \mid Y = y)|$.

When *GLS* holds, $\Delta_{\mathrm{CE}}(\widehat{Y})$ is equal to 0. We now give an upper bound on the error gap between source and target, which can also be used to obtain a generalization upper bound on the target risk.

**Theorem 3.1.** (Error Decomposition Theorem) For any classifier $\widehat{Y} = (h \circ g)(X)$,

$$|\varepsilon_S(h \circ g) - \varepsilon_T(h \circ g)| \leq \|\mathcal{D}_S^Y - \mathcal{D}_T^Y\|_1 \cdot \mathrm{BER}_{\mathcal{D}_S}(\widehat{Y} \parallel Y) + 2(k-1)\Delta_{\mathrm{CE}}(\widehat{Y}),$$

where $\|\mathcal{D}_S^Y - \mathcal{D}_T^Y\|_1 := \sum_{i=1}^k |\mathcal{D}_S(Y = i) - \mathcal{D}_T(Y = i)|$ is the $L_1$ distance between $\mathcal{D}_S^Y$ and $\mathcal{D}_T^Y$.

**Remark** The upper bound in Theorem 3.1 provides a way to decompose the error gap between source and target domains. It also immediately gives a generalization bound on the target risk $\varepsilon_T(h \circ g)$. The bound contains two terms. The first contains $\|\mathcal{D}_S^Y - \mathcal{D}_T^Y\|_1$, which measures the distance between the marginal label distributions across domains and is a constant that only depends on the adaptation problem itself, and BER, a reweighted classification performance on the source domain. The second is $\Delta_{\mathrm{CE}}(\widehat{Y})$ measures the distance between the family of conditional distributions $\widehat{Y} \mid Y$. In other words, the bound is *oblivious* to the optimal labeling functions in feature space. This is in sharp contrast with upper bounds from previous work [7, Theorem 2], [62, Theorem 4.1], which essentially decompose the error gap in terms of the distance between the marginal feature distributions ($\mathcal{D}_S^Z$, $\mathcal{D}_T^Z$) and the optimal labeling functions ($f_S^Z$, $f_T^Z$). Because the optimal labeling function in feature space depends on $Z$ and is unknown in practice, such decomposition is not very informative. As a comparison, Theorem 3.1 provides a decomposition orthogonal to previous results and does not require knowledge about unknown optimal labeling functions in feature space.

Notably, the balanced error rate, $\mathrm{BER}_{\mathcal{D}_S}(\widehat{Y} \parallel Y)$, only depends on samples from the source domain and can be minimized. Furthermore, using a data-processing argument, the conditional error gap $\Delta_{\mathrm{CE}}(\widehat{Y})$ can be minimized by aligning the conditional feature distributions across domains. Putting everything together, the result suggests that, to minimize the error gap, it suffices to align the conditional distributions $Z \mid Y = y$ while simultaneously minimizing the balanced error rate. In fact, under the assumption that the conditional distributions are perfectly aligned (i.e., under *GLS*), we can prove a stronger result, guaranteeing that the joint error is small:

**Theorem 3.2.** If $Z = g(X)$ satisfies *GLS*, then for any $h : \mathcal{Z} \to \mathcal{Y}$ and letting $\widehat{Y} = h(Z)$ be the predictor, we have $\varepsilon_S(\widehat{Y}) + \varepsilon_T(\widehat{Y}) \leq 2\mathrm{BER}_{\mathcal{D}_S}(\widehat{Y} \parallel Y)$.

## 3.3 Conditions for Generalized Label Shift

The main difficulty in applying a bound minimization algorithm inspired by Theorem 3.1 is that we do not have labels from the target domain in UDA, so we cannot directly align the conditional label distributions. By using relative class weights between domains, we can provide a necessary condition for *GLS* that bypasses an explicit alignment of the conditional feature distributions.

**Definition 3.4.** Assuming $\mathcal{D}_S(Y = y) > 0, \forall y \in \mathcal{Y}$, we let $\mathbf{w} \in \mathbb{R}^k$ denote the importance weights of the target and source label distributions:

$$\mathbf{w}_y := \frac{\mathcal{D}_T(Y = y)}{\mathcal{D}_S(Y = y)}, \quad \forall y \in \mathcal{Y}. \tag{4}$$

**Lemma 3.1.** (Necessary condition for *GLS*) If $Z = g(X)$ satisfies *GLS*, then $\mathcal{D}_T(\widetilde{Z}) = \sum_{y \in \mathcal{Y}} \mathbf{w}_y \cdot \mathcal{D}_S(\widetilde{Z}, Y = y) =: \mathcal{D}_S^{\mathbf{w}}(\widetilde{Z})$ where $\widetilde{Z}$ verifies either $\widetilde{Z} = Z$ or $\widetilde{Z} = \widehat{Y} \otimes Z$.

Compared to previous work that attempts to align $\mathcal{D}_T(Z)$ with $\mathcal{D}_S(Z)$ (using adversarial discriminators [20] or maximum mean discrepancy (MMD) [34]) or $\mathcal{D}_T(\widehat{Y} \otimes Z)$ with $\mathcal{D}_S(\widehat{Y} \otimes Z)$ [37], Lemma 3.1 suggests to instead align $\mathcal{D}_T(\widetilde{Z})$ with the *reweighted* marginal distribution $\mathcal{D}_S^{\mathbf{w}}(\widetilde{Z})$. Reciprocally, the following two theorems give sufficient conditions to know when perfectly aligned target feature distribution and reweighted source feature distribution imply *GLS*:

**Theorem 3.3.** (Clustering structure implies sufficiency) Let $Z = g(X)$ such that $\mathcal{D}_T(Z) = \mathcal{D}_S^{\mathbf{w}}(Z)$. Assume $\mathcal{D}_T(Y = y) > 0, \forall y \in \mathcal{Y}$. If there exists a partition of $\mathcal{Z} = \cup_{y \in \mathcal{Y}} \mathcal{Z}_y$ such that $\forall y \in \mathcal{Y}$, $\mathcal{D}_S(Z \in \mathcal{Z}_y \mid Y = y) = \mathcal{D}_T(Z \in \mathcal{Z}_y \mid Y = y) = 1$, then $Z = g(X)$ satisfies *GLS*.

**Remark**    Theorem 3.3 shows that if there exists a partition of the feature space such that instances with the same label are within the same component, then aligning the target feature distribution with the reweighted source feature distribution implies *GLS*. While this clustering assumption may seem strong, it is consistent with the goal of reducing classification error: if such a clustering exists, then there also exists a perfect predictor based on the feature $Z = g(X)$, i.e., the cluster index.

**Theorem 3.4.** Let $\widehat{Y} = h(Z)$, $\gamma := \min_{y \in \mathcal{Y}} \mathcal{D}_T(Y = y)$ and $\mathbf{w}_M := \max_{y \in \mathcal{Y}} \mathbf{w}_y$. For $\widetilde{Z} = Z$ or $\widetilde{Z} = \widehat{Y} \otimes Z$, we have:

$$\max_{y \in \mathcal{Y}} d_{\text{TV}}(\mathcal{D}_S(Z \mid Y = y), \mathcal{D}_T(Z \mid Y = y)) \leq \frac{\mathbf{w}_M \varepsilon_S(\widehat{Y}) + \varepsilon_T(\widehat{Y}) + \sqrt{8 D_{\text{JS}}(\mathcal{D}_S^{\mathbf{w}}(\widetilde{Z}) \| \mathcal{D}_T(\widetilde{Z}))}}{\gamma}.$$

Theorem 3.4 confirms that matching $\mathcal{D}_T(\widetilde{Z})$ with $\mathcal{D}_S^{\mathbf{w}}(\widetilde{Z})$ is the proper objective in the context of mismatched label distributions. It shows that, for matched feature distributions and a source error equal to zero, successful domain adaptation (*i.e.* a target error equal to zero) implies that *GLS* holds. Combined with Theorem 3.2, we even get equivalence between the two.

**Remark**    Thm. 3.4 extends Thm. 3.3 by incorporating the clustering assumption in the joint error achievable by a classifier $\widehat{Y}$ based on a fixed $Z$. In particular, if the clustering structure holds, the joint error is 0 for an appropriate $h$, and aligning the reweighted feature distributions implies *GLS*.

### 3.4    Estimating the Importance Weights w

Inspired by the moment matching technique to estimate $\mathbf{w}$ under label shift from Lipton et al. [31], we propose a method to get $\mathbf{w}$ under *GLS* by solving a quadratic program (QP).

**Definition 3.5.** We let $\mathbf{C} \in \mathbb{R}^{|\mathcal{Y}| \times |\mathcal{Y}|}$ denote the confusion matrix of the classifier on the source domain and $\mu \in \mathbb{R}^{|\mathcal{Y}|}$ the distribution of predictions on the target one, $\forall y, y' \in \mathcal{Y}$:

$$\mathbf{C}_{y,y'} := \mathcal{D}_S(\widehat{Y} = y, Y = y'), \qquad \mu_y := \mathcal{D}_T(\widehat{Y} = y).$$

**Lemma 3.2.** If *GLS* is verified, and if the confusion matrix $\mathbf{C}$ is invertible, then $\mathbf{w} = \mathbf{C}^{-1}\mu$.

The key insight from Lemma 3.2 is that, to estimate the importance vector $\mathbf{w}$ under *GLS*, we do not need access to labels from the target domain. However, matrix inversion is notoriously numerically unstable, especially with finite sample estimates $\widehat{\mathbf{C}}$ and $\widehat{\mu}$ of $\mathbf{C}$ and $\mu$. We propose to solve instead the following QP (written as $QP(\widehat{\mathbf{C}}, \widehat{\mu})$), whose solution will be consistent if $\widehat{\mathbf{C}} \to \mathbf{C}$ and $\widehat{\mu} \to \mu$:

$$\underset{\mathbf{w}}{\text{minimize}} \quad \frac{1}{2} \|\widehat{\mu} - \widehat{\mathbf{C}}\mathbf{w}\|_2^2, \qquad \text{subject to} \quad \mathbf{w} \geq 0, \, \mathbf{w}^T \mathcal{D}_S(Y) = 1. \tag{5}$$

The above program (5) can be efficiently solved in time $O(|\mathcal{Y}|^3)$, with $|\mathcal{Y}|$ small and constant; and by construction, its solution is element-wise non-negative, even with limited amounts of data to estimate $\mathbf{C}$ and $\mu$.

---
**Algorithm 1** Importance-Weighted Domain Adaptation
---
1: **Input:** source and target data $(x_S, y_S)$, $x_T$; $g_\theta$, $h_\phi$ and $d_\psi$; epochs $E$, batches per epoch $B$
2: Initialize $\mathbf{w}_1 = 1$
3: **for** $t = 1$ **to** $E$ **do**
4:     Initialize $\hat{\mathbf{C}} = 0$, $\hat{\boldsymbol{\mu}} = 0$
5:     **for** $b = 1$ **to** $B$ **do**
6:         Sample batches $(x_S^i, y_S^i)$ and $(x_T^i)$ of size s
7:         Maximize $\mathcal{L}_{DA}^{\mathbf{w}_t}$ w.r.t. $\theta$, minimize $\mathcal{L}_{DA}^{\mathbf{w}_t}$ w.r.t. $\psi$ and minimize $\mathcal{L}_C^{\mathbf{w}_t}$ w.r.t. $\theta$ and $\phi$
8:         **for** $i = 1$ **to** $s$ **do**
9:             $\hat{\mathbf{C}}_{\cdot y_S^i} \leftarrow \hat{\mathbf{C}}_{\cdot y_S^i} + h_\phi(g_\theta(x_S^i))$ ($y_S^i$-th column)    and    $\hat{\boldsymbol{\mu}} \leftarrow \hat{\boldsymbol{\mu}} + h_\phi(g_\theta(x_T^i))$
10:    $\hat{\mathbf{C}} \leftarrow \hat{\mathbf{C}}/sB$ and $\hat{\boldsymbol{\mu}} \leftarrow \hat{\boldsymbol{\mu}}/sB$;    then    $\mathbf{w}_{t+1} = \lambda \cdot QP(\hat{\mathbf{C}}, \hat{\boldsymbol{\mu}}) + (1 - \lambda)\mathbf{w}_t$
---

**Lemma 3.3.** If the source error $\varepsilon_S(h \circ g)$ is zero and the source and target marginals verify $D_{\text{JS}}(\mathcal{D}_S^{\tilde{\mathbf{w}}}(Z), \mathcal{D}_T(Z)) = 0$, then the estimated weight vector $\mathbf{w}$ is equal to $\tilde{\mathbf{w}}$.

Lemma 3.3 shows that the weight estimation is stable once the DA losses have converged, but it does not imply convergence to the true weights (see Sec. 4.2 and App. **??** for more details).

### 3.5  $\mathcal{F}$-IPM for Distributional Alignment

To align the target feature distribution and the reweighted source feature distribution as suggested by Lemma 3.1, we now provide a general framework using the integral probability metric [40, IPM].

**Definition 3.6.** With $\mathcal{F}$ a set of real-valued functions, the $\mathcal{F}$-IPM between distributions $\mathcal{D}$ and $\mathcal{D}'$ is

$$d_\mathcal{F}(\mathcal{D}, \mathcal{D}') := \sup_{f \in \mathcal{F}} |\mathbb{E}_{X \sim \mathcal{D}}[f(X)] - \mathbb{E}_{X \sim \mathcal{D}'}[f(X)]|. \tag{6}$$

By approximating any function class $\mathcal{F}$ using parametrized models, e.g., neural networks, we obtain a general framework for domain adaptation by aligning reweighted source feature distribution and target feature distribution, i.e. by minimizing $d_\mathcal{F}(\mathcal{D}_T(\tilde{Z}), \mathcal{D}_S^{\mathbf{w}}(\tilde{Z}))$. Below, by instantiating $\mathcal{F}$ to be the set of bounded norm functions in a RKHS $\mathcal{H}$ [25], we obtain maximum mean discrepancy methods, leading to IWJAN (cf. Section 4.1), a variant of JAN [36] for UDA.

## 4  Practical Implementation

### 4.1  Algorithms

The sections above suggest simple algorithms based on representation learning: (i) estimate $\mathbf{w}$ on the fly during training, (ii) align the feature distributions $\tilde{Z}$ of the target domain with the reweighted feature distribution of the source domain and, (iii) minimize the balanced error rate. Overall, we present the pseudocode of our algorithm in Alg. 1.

To compute $\mathbf{w}$, we build estimators $\hat{\mathbf{C}}$ and $\hat{\boldsymbol{\mu}}$ of $\mathbf{C}$ and $\boldsymbol{\mu}$ by averaging during each epoch the predictions of the classifier on the source (per true class) and target (overall). This corresponds to the inner-most loop of Algorithm 1 (lines 8-9). At epoch end, $\mathbf{w}$ is updated (line 10), and the estimators reset to 0. We have found empirically that using an exponential moving average of $\mathbf{w}$ performs better. Our results all use a factor $\lambda = 0.5$. We also note that Alg. 1 implies a minimal computational overhead (see App. **??** for details): in practice our algorithms run as fast as their base versions.

Using $\mathbf{w}$, we can define our first algorithm, *Importance-Weighted Domain Adversarial Network* (IWDAN), that aligns $\mathcal{D}_S^{\mathbf{w}}(Z)$ and $\mathcal{D}_T(Z)$) using a discriminator. To that end, we modify the DANN losses $\mathcal{L}_{DA}$ and $\mathcal{L}_C$ as follows. For batches $(x_S^i, y_S^i)$ and $(x_T^i)$ of size $s$, the weighted DA loss is:

$$\mathcal{L}_{DA}^{\mathbf{w}}(x_S^i, y_S^i, x_T^i; \theta, \psi) = -\frac{1}{s}\sum_{i=1}^{s}\mathbf{w}_{y_S^i}\log(d_\psi(g_\theta(x_S^i))) + \log(1 - d_\psi(g_\theta(x_T^i))). \tag{7}$$

We verify in App. **??**, that the standard ADA framework applied to $\mathcal{L}_{DA}^{\mathbf{w}}$ indeed minimizes $D_{\text{JS}}(\mathcal{D}_S^{\mathbf{w}}(Z)\|\mathcal{D}_T(Z))$. Our second algorithm, *Importance-Weighted Joint Adaptation Networks*

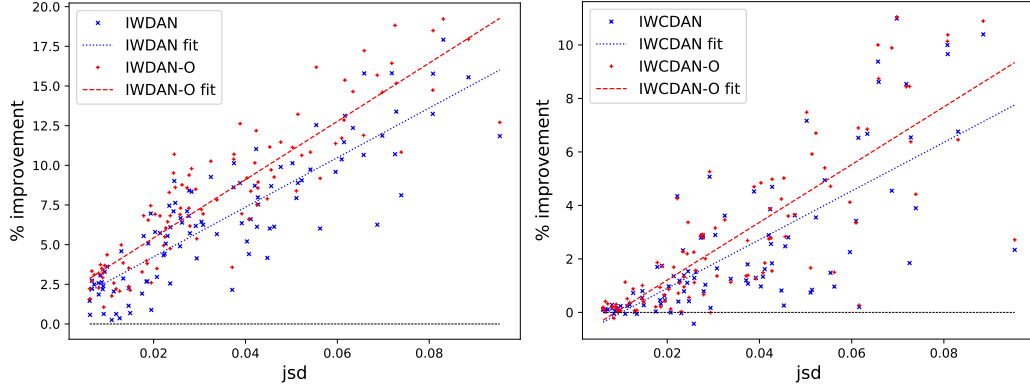

Figure 1: Gains of our algorithms vs their base versions (the horizontal grey line) for 100 tasks. The $x$-axis is $D_{\text{JS}}(\mathcal{D}_S^Y, \mathcal{D}_T^Y)$. The mean improvements for IWDAN and IWDAN-O (resp. IWCDAN and IWCDAN-O) are 6.55% and 8.14% (resp. 2.25% and 2.81%).

(IWJAN) is based on JAN [36] and follows the reweighting principle described in Section 3.5 with $\mathcal{F}$ a learnt RKHS (the exact JAN and IWJAN losses are specified in App. **??**). Finally, our third algorithm is *Importance-Weighted Conditional Domain Adversarial Network* (IWCDAN). It matches $\mathcal{D}_S^{\mathbf{w}}(\hat{Y} \otimes Z)$ with $\mathcal{D}_T(\hat{Y} \otimes Z)$ by replacing the standard adversarial loss in CDAN with Eq. 7, where $d_\psi$ takes as input $(h_\phi \circ g_\theta) \otimes g_\theta$ instead of $g_\theta$. The classifier loss for our three variants is:

$$\mathcal{L}_C^{\mathbf{w}}(x_S^i, y_S^i; \theta, \phi) = -\frac{1}{s} \sum_{i=1}^s \frac{1}{k \cdot \mathcal{D}_S(Y = y)} \log(h_\phi(g_\theta(x_S^i))_{y_S^i}). \tag{8}$$

This reweighting is suggested by our theoretical analysis from Section 3, where we seek to minimize the balanced error rate $\text{BER}_{\mathcal{D}_S}(\hat{Y} \parallel Y)$. We also define oracle versions, IWDAN-O, IWJAN-O and IWCDAN-O, where the weights $\mathbf{w}$ are the true weights. It gives an idealistic version of the reweighting method, and allows to assess the soundness of $GLS$. IWDAN, IWJAN and IWCDAN are Alg. 1 with their respective loss functions, the oracle versions use the true weights instead of $\mathbf{w}_t$.

### 4.2 Experiments

We apply our three base algorithms, their importance weighted versions, and the oracles to 4 standard DA datasets generating 21 tasks: Digits (MNIST $\leftrightarrow$ USPS [18, 29]), Visda [51], Office-31 [45] and Office-Home [50]. All values are averages over 5 runs of the best test accuracy throughout training (evaluated at the end of each epoch). We used that value for fairness with respect to the baselines (as shown in the left panel of Figure 2, the performance of DANN decreases as training progresses, due to the inappropriate matching of representations showcased in Theorem 2.1). For full details, see App. **??** and **??**.

**Performance vs $D_{\text{JS}}$** We artificially generate 100 tasks from MNIST and USPS by considering various random subsets of the classes in either the source or target domain (see Appendix **??** for details). These 100 DA tasks have a $D_{\text{JS}}(\mathcal{D}_S^Y, \mathcal{D}_T^Y)$ varying between 0 and 0.1. Applying IWDAN and IWCDAN results in Fig. 1. We see a clear correlation between the improvements provided by our algorithms and $D_{\text{JS}}(\mathcal{D}_S^Y, \mathcal{D}_T^Y)$, which is well aligned with Theorem 2.1. Moreover, IWDAN outperfoms DANN on the 100 tasks and IWCDAN bests CDAN on 94. Even on small divergences, our algorithms do not suffer compared to their base versions.

**Original Datasets** The average results on each dataset are shown in Table 2 (see App.**??** for the per-task breakdown). IWDAN outperforms the basic algorithm DANN by 1.75%, 1.64%, 1.16% and 2.65% on the Digits, Visda, Office-31 and Office-Home tasks respectively. Gains for IWCDAN are more limited, but still present: 0.18%, 0.89%, 0.07% and 1.07% respectively. This might be explained by the fact that CDAN enforces a weak form of $GLS$ (App. **??**). Gains for JAN are 0.58%, 0.19% and 0.19%. We also show the fraction of times (over all seeds and tasks) our variants outperform the original algorithms. Even for small gains, the variants provide consistent improvements. Additionally, the oracle versions show larger improvements, which strongly supports enforcing $GLS$.

Table 2: Average results on the various domains (Digits has 2 tasks, Visda 1, Office-31 6 and Office-Home 12). The prefix $s$ denotes the experiment where the source domain is subsampled to increase $D_{\text{JS}}(\mathcal{D}_S^Y, \mathcal{D}_T^Y)$. Each number is a mean over 5 seeds, the subscript denotes the fraction of times (out of $5\ seeds \times \#tasks$) our algorithms outperform their base versions. JAN is not available on Digits.

| METHOD | DIGITS | sDIGITS | VISDA | sVISDA | O-31 | sO-31 | O-H | sO-H |
|---|---|---|---|---|---|---|---|---|
| NO DA | 77.17 | 75.67 | 48.39 | 49.02 | 77.81 | 75.72 | 56.39 | 51.34 |
| DANN | 93.15 | 83.24 | 61.88 | 52.85 | 82.74 | 76.17 | 59.62 | 51.83 |
| IWDAN | $\mathbf{94.90}_{100\%}$ | $\mathbf{92.54}_{100\%}$ | $\mathbf{63.52}_{100\%}$ | $\mathbf{60.18}_{100\%}$ | $\mathbf{83.90}_{87\%}$ | $\mathbf{82.60}_{100\%}$ | $\mathbf{62.27}_{97\%}$ | $\mathbf{57.61}_{100\%}$ |
| IWDAN-O | $95.27_{100\%}$ | $94.46_{100\%}$ | $64.19_{100\%}$ | $62.10_{100\%}$ | $85.33_{97\%}$ | $84.41_{100\%}$ | $64.68_{100\%}$ | $60.87_{100\%}$ |
| CDAN | 95.72 | 88.23 | 65.60 | 60.19 | 87.23 | 81.62 | 64.59 | 56.25 |
| IWCDAN | $\mathbf{95.90}_{80\%}$ | $\mathbf{93.22}_{100\%}$ | $\mathbf{66.49}_{60\%}$ | $\mathbf{65.83}_{100\%}$ | $\mathbf{87.30}_{73\%}$ | $\mathbf{83.88}_{100\%}$ | $\mathbf{65.66}_{70\%}$ | $\mathbf{61.24}_{100\%}$ |
| IWCDAN-O | $95.85_{90\%}$ | $94.81_{100\%}$ | $68.15_{100\%}$ | $66.85_{100\%}$ | $88.14_{90\%}$ | $85.47_{100\%}$ | $67.64_{98\%}$ | $63.73_{100\%}$ |
| JAN | N/A | N/A | 56.98 | 50.64 | 85.13 | 78.21 | 59.59 | 53.94 |
| IWJAN | N/A | N/A | $\mathbf{57.56}_{100\%}$ | $\mathbf{57.12}_{100\%}$ | $\mathbf{85.32}_{60\%}$ | $\mathbf{82.61}_{97\%}$ | $\mathbf{59.78}_{63\%}$ | $\mathbf{55.89}_{100\%}$ |
| IWJAN-O | N/A | N/A | $61.48_{100\%}$ | $61.30_{100\%}$ | $87.14_{100\%}$ | $86.24_{100\%}$ | $60.73_{92\%}$ | $57.36_{100\%}$ |

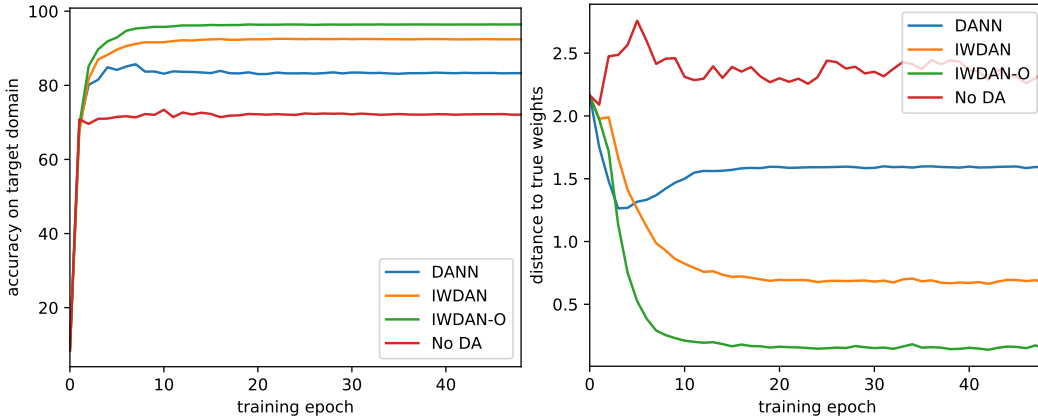

Figure 2: *Left* Accuracy on sDigits. *Right* Euclidian distance between estimated and true weights.

**Subsampled datasets** The original datasets have fairly balanced classes, making the JSD between source and target label distributions $D_{\text{JS}}(\mathcal{D}_S^Y \parallel \mathcal{D}_T^Y)$ rather small (Tables **??**, **??** and **??** in App. **??**). To evaluate our algorithms on larger divergences, we arbitrarily modify the source domains above by considering only 30% of the samples coming from the first half of their classes. This results in larger divergences (Tables **??**, **??** and **??**). Performance is shown in Table 2 (datasets prefixed by s). For IWDAN, we see gains of 9.3%, 7.33%, 6.43% and 5.58% on the digits, Visda, Office-31 and Office-Home datasets respectively. For IWCDAN, improvements are 4.99%, 5.64%, 2.26% and 4.99%, and IWJAN shows gains of 6.48%, 4.40% and 1.95%. Moreover, on all seeds and tasks but one, our variants outperform their base versions.

**Importance weights** While our method demonstrates gains empirically, Lemma 3.2 does not guarantee convergence of **w** to the true weights. In Fig. 2, we show the test accuracy and distance between estimated and true weights during training on sDigits. We see that DANN's performance gets worse after a few epoch, as predicted by Theorem 2.1. The representation matching objective collapses classes that are over-represented in the target domain on the under-represented ones (see App. **??**). This phenomenon does not occur for IWDAN and IWDAN-O. Both monotonously improve in accuracy and estimation (see Lemma 3.3 and App. **??** for more details). We also observe that IWDAN's weights do not converge perfectly. This suggests that fine-tuning $\lambda$ (we used $\lambda = 0.5$ in all our experiments for simplicity) or updating **w** more or less often could lead to better performance.

**Ablation Study** Our algorithms have two components, a weighted adversarial loss $\mathcal{L}_{DA}^{\mathbf{w}}$ and a weighted classification loss $\mathcal{L}_C^{\mathbf{w}}$. In Table 3, we augment DANN and CDAN using those losses separately (with the true weights). We observe that DANN benefits essentially from the reweighting of its adversarial loss $\mathcal{L}_{DA}^{\mathbf{w}}$, the classification loss has little effect. For CDAN, gains are essentially seen on the subsampled datasets. Both losses help, with a +2% extra gain for $\mathcal{L}_{DA}^{\mathbf{w}}$.

Table 3: Ablation study on the Digits tasks.

| Method | Digits | sDigits | Method | Digits | sDigits |
|---|---|---|---|---|---|
| DANN | 93.15 | 83.24 | CDAN | 95.72 | 88.23 |
| DANN + $\mathcal{L}_C^{\mathbf{w}}$ | 93.27 | 84.52 | CDAN + $\mathcal{L}_C^{\mathbf{w}}$ | 95.65 | 91.01 |
| DANN + $\mathcal{L}_{DA}^{\mathbf{w}}$ | **95.31** | **94.41** | CDAN + $\mathcal{L}_{DA}^{\mathbf{w}}$ | 95.42 | 93.18 |
| IWDAN-O | **95.27** | **94.46** | IWCDAN-O | **95.85** | **94.81** |

## 5 Related Work

Covariate shift has been studied and used in many adaptation algorithms [1, 3, 24, 28, 44, 49, 61]. While less known, label shift has also been tackled from various angles over the years: applying EM to learn $\mathcal{D}_T^Y$ [13], placing a prior on the label distribution [48], using kernel mean matching [19, 42, 57], etc. Schölkopf et al. [46] cast the problem in a causal/anti-causal perspective corresponding to covariate/label shift. That perspective was then further developed [4, 21, 31, 57]. Numerous domain adaptation methods rely on learning invariant representations, and minimize various metrics on the marginal feature distributions: total variation or equivalently $D_{\text{JS}}$ [20, 32, 49, 60], maximum mean discrepancy [25, 33–36], Wasserstein distance [14–16, 30, 47], etc. Other noteworthy DA methods use reconstruction losses and cycle-consistency to learn transferable classifiers [27, 52, 63]. Recently, Liu et al. [32] have introduced Transferable Adversarial Training (TAT), where transferable examples are generated to fill the gap in feature space between source and target domains, the datasets is then augmented with those samples. Applying our method to TAT is a future research direction.

Other relevant settings include partial ADA, i.e. UDA when target labels are a strict subset of the source labels / some components of $\mathbf{w}$ are 0 [10–12]. Multi-domain adaptation, where multiple source or target domains are given, is also very studied [17, 26, 38, 41, 43, 59]. Recently, Binkowski et al. [8] study sample reweighting in the domain transfer to handle mass shifts between distributions.

Prior work on combining importance weight in domain-invariant representation learning also exists in the setting of partial DA [56]. However, the importance ratio in these works is defined over the features $Z$, rather than the class label $Y$. Compared to our method, this is both statistically inefficient and computationally expensive, since the feature space $\mathcal{Z}$ is often a high-dimensional continuous space, whereas the label space $\mathcal{Y}$ only contains a finite number ($k$) of distinct labels. In a separate work, Yan et al. [54] proposed a weighted MMD distance to handle target shift in UDA. However, their weights are estimated based on pseudo-labels obtained from the learned classifier, hence it is not clear whether the pseudo-labels provide accurate estimation of the importance weights even in simple settings. As a comparison, under $GLS$, we show that our weight estimation by solving a quadratic program converges asymptotically.

## 6 Conclusion and Future Work

We have introduced the generalized label shift assumption, $GLS$, and theoretically-grounded variations of existing algorithms to handle mismatched label distributions. On tasks from classic benchmarks as well as artificial ones, our algorithms consistently outperform their base versions. The gains, as expected theoretically, correlate well with the JSD between label distributions across domains. In real-world applications, the JSD is unknown, and might be larger than in ML datasets where classes are often purposely balanced. Being simple to implement and adding barely any computational cost, the robustness of our method to mismatched label distributions makes it very relevant to such applications.

**Extensions** The framework we define in this paper relies on appropriately reweighting the domain adversarial losses. It can be straightforwardly applied to settings where multiple source and/or target domains are used, by simply maintaining one importance weights vector $\mathbf{w}$ for each source/target pair [43, 59]. In particular, label shift could explain the observation from Zhao et al. [59] that too many source domains hurt performance, and our framework might alleviate the issue. One can also think of settings (e.g. semi-supervised domain adaptation) where estimations of $\mathcal{D}_T^Y$ can be obtained via other means. A more challenging but also more interesting future direction is to extend our framework to *domain generalization*, where the learner has access to multiple labeled source domains but no access to (even unlabelled) data from the target domain.

## Acknowledgements

The authors thank Romain Laroche and Alessandro Sordoni for useful feedback and helpful discussions. HZ and GG would like to acknowledge support from the DARPA XAI project, contract #FA87501720152 and a Nvidia GPU grant. YW would like acknowledge partial support from NSF Award #2029626, a start-up grant from UCSB Department of Computer Science, as well as generous gifts from Amazon, Adobe, Google and NEC Labs.

## Broader Impact

Our work focuses on domain adaptation and attempts to properly handle mismatches in the label distributions between the source and target domains. Domain Adaptation as a whole aims at transferring knowledge gained from a certain domain (or data distribution) to another one. It can potentially be used in a variety of decision making systems, such as spam filters, machine translation, etc.. One can also potentially think of much more sensitive applications such as recidivism prediction, or loan approvals.

While it is unclear to us to what extent DA is currently applied, or how it will be applied in the future, the bias formalized in Th. 2.1 and verified in Table **??** demonstrates that imbalances between classes will result in poor transfer performance of standard ADA methods on a subset of them, which is without a doubt a source of potential inequalities. Our method is actually aimed at counter-balancing the effect of such imbalances. As shown in our empirical results (for instance Table **??**) it is rather successful at it, especially on significant shifts. This makes us rather confident in the algorithm's ability to mitigate potential effects of biases in the datasets. On the downside, failure in the weight estimation of some classes might result in poor performance on those. However, we have not observed, in any of our experiments, our method performing significantly worse than its base version. Finally, our method is a variation over existing deep learning algorithms. As such, it carries with it the uncertainties associated to deep learning models, in particular a lack of interpretability and of formal convergence guarantees.

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
