[Supplementary Material]

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

## Footnotes

[2]We manually rejected some samples to guarantee a rather uniform set of divergences.

[3]https://github.com/thuml/CDAN/tree/master/pytorch

[4]https://github.com/thuml/Xlearn/tree/master/pytorch

[5]http://cvxopt.org/

[6]It does not reach it as the learning rate is decayed to 0.

[7]In particular at initialization, one class usually dominates the others.

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

# A Omitted Proofs

In this section, we provide the theoretical material that completes the main text.

## A.1 Definition

**Definition A.1.** Let us recall that for two distributions $\mathcal{D}$ and $\mathcal{D}'$, the Jensen-Shannon (JSD) divergence $D_{\mathrm{JS}}(\mathcal{D} \parallel \mathcal{D}')$ is defined as:

$$D_{\mathrm{JS}}(\mathcal{D} \parallel \mathcal{D}') := \frac{1}{2} D_{\mathrm{KL}}(\mathcal{D} \parallel \mathcal{D}_M) + \frac{1}{2} D_{\mathrm{KL}}(\mathcal{D}' \parallel \mathcal{D}_M),$$

where $D_{\mathrm{KL}}(\cdot \parallel \cdot)$ is the Kullback–Leibler (KL) divergence and $\mathcal{D}_M := (\mathcal{D} + \mathcal{D}')/2$.

## A.2 Consistency of the Weighted Domain Adaptation Loss (7)

For the sake of conciseness, we verify here that the domain adaptation training objective does lead to minimizing the Jensen-Shannon divergence between the weighted feature distribution of the source domain and the feature distribution of the target domain.

**Lemma A.1.** Let $p(x, y)$ and $q(x)$ be two density distributions, and $w(y)$ be a positive function such that $\int p(y)w(y)dy = 1$. Let $p^w(x) = \int p(x, y)w(y)dy$ denote the $w$-reweighted marginal distribution of $x$ under $p$. The minimum value of

$$I(d) := \mathbb{E}_{(x,y)\sim p, x'\sim q}[-w(y)\log(d(x)) - \log(1 - d(x'))]$$

is $\log(4) - 2D_{\mathrm{JS}}(p^w(x) \parallel q(x))$, and is attained for $d^*(x) = \frac{p^w(x)}{p^w(x)+q(x)}$.

*Proof.* We see that:

$$I(d) = -\iiint [w(y)\log(d(x)) + \log(1 - d(x'))]p(x,y)q(x')dxdx'dy \tag{9}$$

$$= -\int [\int w(y)p(x,y)dy]\log(d(x)) + q(x)\log(1 - d(x))dx \tag{10}$$

$$= -\int p^w(x)\log(d(x)) + q(x)\log(1 - d(x))dx. \tag{11}$$

From the last line, we follow the exact method from Goodfellow et al. [25] to see that point-wise in $x$ the minimum is attained for $d^*(x) = \frac{p^w(x)}{p^w(x)+q(x)}$ and that $I(d^*) = \log(4) - 2D_{\mathrm{JS}}(p^w(x) \parallel q(x))$. ∎

Applying Lemma A.1 to $\mathcal{D}_S(Z, Y)$ and $\mathcal{D}_T(Z)$ proves that the domain adaptation objective leads to minimizing $D_{\mathrm{JS}}(\mathcal{D}_S^w(Z) \parallel \mathcal{D}_T(Z))$.

## A.3 $k$-class information-theoretic lower bound

In this section, we prove Theorem 2.1 that extends previous result to the general $k$-class classification problem.

**Theorem 2.1.** Assuming that $D_{\mathrm{JS}}(\mathcal{D}_S^Y \parallel \mathcal{D}_T^Y) \geq D_{\mathrm{JS}}(\mathcal{D}_S^{\widetilde{Z}} \parallel \mathcal{D}_T^{\widetilde{Z}})$, then:

$$\varepsilon_S(h \circ g) + \varepsilon_T(h \circ g) \geq \frac{1}{2}\left(\sqrt{D_{\mathrm{JS}}(\mathcal{D}_S^Y \parallel \mathcal{D}_T^Y)} - \sqrt{D_{\mathrm{JS}}(\mathcal{D}_S^{\widetilde{Z}} \parallel \mathcal{D}_T^{\widetilde{Z}})}\right)^2.$$

*Proof.* We essentially follow the proof from Zhao et al. [66], except for Lemmas 4.6 that needs to be adapted to the CDAN framework and Lemma 4.7 to $k$-class classification.

Lemma 4.6 from Zhao et al. [66] states that $D_{\mathrm{JS}}(\mathcal{D}_S^{\widehat{Y}}, \mathcal{D}_T^{\widehat{Y}}) \leq D_{\mathrm{JS}}(\mathcal{D}_S^Z, \mathcal{D}_T^Z)$, which covers the case $\widetilde{Z} = Z$.

When $\widetilde{Z} = \widehat{Y} \otimes Z$, let us first recall that we assume $h$ or equivalently $\widehat{Y}$ to be a one-hot prediction of the class. We have the following Markov chain:

$$X \xrightarrow{g} Z \xrightarrow{\tilde{h}} \widetilde{Z} \xrightarrow{l} \widehat{Y},$$

where $\tilde{h}(z) = h(z) \otimes z$ and $l : \mathcal{Y} \otimes \mathcal{Z} \to \mathcal{Y}$ returns the index of the non-zero block in $\tilde{h}(z)$. There is only one such block since $h$ is a one-hot, and its index corresponds to the class predicted by $h$. Given the definition of $l$, we clearly see that $\widehat{Y}$ is independent of $X$ knowing $\widetilde{Z}$. We can now apply the same proof than in Zhao et al. [66] to conclude that:

$$D_{\text{JS}}(\mathcal{D}_S^{\widehat{Y}}, \mathcal{D}_T^{\widehat{Y}}) \leq D_{\text{JS}}(\mathcal{D}_S^{\widetilde{Z}}, \mathcal{D}_T^{\widetilde{Z}}). \tag{12}$$

It essentially boils down to a data-processing argument: the discrimination distance between two distributions cannot increase after the same (possibly stochastic) channel (kernel) is applied to both. Here, the channel corresponds to the (potentially randomized) function $l$.

**Remark**   Additionally, we note that the above inequality holds for *any* $\widetilde{Z}$ such that $\widehat{Y} = l(\widetilde{Z})$ for a (potentially randomized) function l. This covers any and all potential combinations of representations at various layers of the deep net, including the last layer (which corresponds to its predictions $\widehat{Y}$).

Let us move to the second part of the proof. We wish to show that $D_{\text{JS}}(\mathcal{D}^Y, \mathcal{D}^{\widehat{Y}}) \leq \varepsilon(h \circ g)$, where $\mathcal{D}$ can be either $\mathcal{D}_S$ or $\mathcal{D}_T$:

$$2D_{\text{JS}}(\mathcal{D}^Y, \mathcal{D}^{\widehat{Y}}) \leq \|\mathcal{D}^Y - \mathcal{D}^{\widehat{Y}}\|_1 \qquad\qquad [34]$$

$$= \sum_{i=1}^{k} |\mathcal{D}(\widehat{Y} = i) - \mathcal{D}(Y = i)|$$

$$= \sum_{i=1}^{k} |\sum_{j=1}^{k} \mathcal{D}(\widehat{Y} = i|Y = j)\mathcal{D}(Y = j) - \mathcal{D}(Y = i)|$$

$$= \sum_{i=1}^{k} |\mathcal{D}(\widehat{Y} = i|Y = i)\mathcal{D}(Y = i) - \mathcal{D}(Y = i) + \sum_{j \neq i} \mathcal{D}(\widehat{Y} = i|Y = j)\mathcal{D}(Y = j)|$$

$$\leq \sum_{i=1}^{k} |\mathcal{D}(\widehat{Y} = i|Y = i) - 1|\mathcal{D}(Y = i) + \sum_{i=1}^{k}\sum_{j \neq i} \mathcal{D}(\widehat{Y} = i|Y = j)\mathcal{D}(Y = j)$$

$$= \sum_{i=1}^{k} \mathcal{D}(\widehat{Y} \neq Y|Y = i)\mathcal{D}(Y = i) + \sum_{j=1}^{k}\sum_{i \neq j} \mathcal{D}(\widehat{Y} = i|Y = j)\mathcal{D}(Y = j)$$

$$= 2\sum_{i=1}^{k} \mathcal{D}(\widehat{Y} \neq Y|Y = i)\mathcal{D}(Y = i) = 2\mathcal{D}(\widehat{Y} \neq Y) = 2\varepsilon(h \circ g). \tag{13}$$

We can now apply the triangular inequality to $\sqrt{D_{\text{JS}}}$, which is a distance metric [21], called the Jensen-Shannon distance. This gives us:

$$\sqrt{D_{\text{JS}}(\mathcal{D}_S^Y, \mathcal{D}_T^Y)} \leq \sqrt{D_{\text{JS}}(\mathcal{D}_S^Y, \mathcal{D}_S^{\widehat{Y}})} + \sqrt{D_{\text{JS}}(\mathcal{D}_S^{\widehat{Y}}, \mathcal{D}_T^{\widehat{Y}})} + \sqrt{D_{\text{JS}}(\mathcal{D}_T^{\widehat{Y}}, \mathcal{D}_T^Y)}$$

$$\leq \sqrt{D_{\text{JS}}(\mathcal{D}_S^Y, \mathcal{D}_S^{\widehat{Y}})} + \sqrt{D_{\text{JS}}(\mathcal{D}_S^{\widetilde{Z}}, \mathcal{D}_T^{\widetilde{Z}})} + \sqrt{D_{\text{JS}}(\mathcal{D}_T^{\widehat{Y}}, \mathcal{D}_T^Y)}$$

$$\leq \sqrt{\varepsilon_S(h \circ g)} + \sqrt{D_{\text{JS}}(\mathcal{D}_S^{\widetilde{Z}}, \mathcal{D}_T^{\widetilde{Z}})} + \sqrt{\varepsilon_T(h \circ g)}.$$

where we used Equation (12) for the second inequality and (13) for the third.

Finally, assuming that $D_{\text{JS}}(\mathcal{D}_S^Y, \mathcal{D}_T^Y) \geq D_{\text{JS}}(\mathcal{D}_S^{\widetilde{Z}}, \mathcal{D}_T^{\widetilde{Z}})$, we get:

$$\left(\sqrt{D_{\text{JS}}(\mathcal{D}_S^Y, \mathcal{D}_T^Y)} - \sqrt{D_{\text{JS}}(\mathcal{D}_S^{\widetilde{Z}}, \mathcal{D}_T^{\widetilde{Z}})}\right)^2 \leq \left(\sqrt{\varepsilon_S(h \circ g)} + \sqrt{\varepsilon_T(h \circ g)}\right)^2 \leq 2\left(\varepsilon_S(h \circ g) + \varepsilon_T(h \circ g)\right).$$

which concludes the proof. ∎

## A.4 Proof of Theorem 3.1

To simplify the notation, we define the error gap $\Delta_\varepsilon(\widehat{Y})$ as follows:

$$\Delta_\varepsilon(\widehat{Y}) := |\varepsilon_S(\widehat{Y}) - \varepsilon_T(\widehat{Y})|.$$

Also, in this case we use $\mathcal{D}_a$, $a \in \{S, T\}$ to mean the source and target distributions respectively. Before we give the proof of Theorem 3.1, we first prove the following two lemmas that will be used in the proof.

**Lemma A.2.** Define $\gamma_{a,j} := \mathcal{D}_a(Y = j), \forall a \in \{S, T\}, \forall j \in [k]$, then $\forall \alpha_j, \beta_j \geq 0$ such that $\alpha_j + \beta_j = 1$, and $\forall i \neq j$, the following upper bound holds:

$$|\gamma_{S,j}\mathcal{D}_S(\widehat{Y} = i \mid Y = j) - \gamma_{T,j}\mathcal{D}_T(\widehat{Y} = i \mid Y = j)| \leq$$
$$|\gamma_{S,j} - \gamma_{T,j}| \cdot \left(\alpha_j\mathcal{D}_S(\widehat{Y} = i \mid Y = j) + \beta_j\mathcal{D}_T(\widehat{Y} = i \mid Y = j)\right) + \gamma_{S,j}\beta_j\Delta_{\mathrm{CE}}(\widehat{Y}) + \gamma_{T,j}\alpha_j\Delta_{\mathrm{CE}}(\widehat{Y}).$$

*Proof.* To make the derivation uncluttered, define $\mathcal{D}_j(\widehat{Y} = i) := \alpha_j\mathcal{D}_S(\widehat{Y} = i \mid Y = j) + \beta_j\mathcal{D}_T(\widehat{Y} = i \mid Y = j)$ to be the mixture conditional probability of $\widehat{Y} = i$ given $Y = j$, where the mixture weight is given by $\alpha_j$ and $\beta_j$. Then in order to prove the upper bound in the lemma, it suffices if we give the desired upper bound for the following term

$$\left| |\gamma_{S,j}\mathcal{D}_S(\widehat{Y} = i \mid Y = j) - \gamma_{T,j}\mathcal{D}_T(\widehat{Y} = i \mid Y = j)| - |(\gamma_{S,j} - \gamma_{T,j})\mathcal{D}_j(\widehat{Y} = i)| \right|$$
$$\leq \left| \left(\gamma_{S,j}\mathcal{D}_S(\widehat{Y} = i \mid Y = j) - \gamma_{T,j}\mathcal{D}_T(\widehat{Y} = i \mid Y = j)\right) - (\gamma_{S,j} - \gamma_{T,j})\mathcal{D}_j(\widehat{Y} = i) \right|$$
$$= \left| \gamma_{S,j}(\mathcal{D}_S(\widehat{Y} = i \mid Y = j) - \mathcal{D}_j(\widehat{Y} = i)) - \gamma_{T,j}(\mathcal{D}_T(\widehat{Y} = i \mid Y = j) - \mathcal{D}_j(\widehat{Y} = i)) \right|,$$

following which we will have:

$$|\gamma_{S,j}\mathcal{D}_S(\widehat{Y} = i \mid Y = j) - \gamma_{T,j}\mathcal{D}_T(\widehat{Y} = i \mid Y = j)| \leq |(\gamma_{S,j} - \gamma_{T,j})\mathcal{D}_j(\widehat{Y} = i)|$$
$$+ \left| \gamma_{S,j}(\mathcal{D}_S(\widehat{Y} = i \mid Y = j) - \mathcal{D}_j(\widehat{Y} = i)) - \gamma_{T,j}(\mathcal{D}_T(\widehat{Y} = i \mid Y = j) - \mathcal{D}_j(\widehat{Y} = i)) \right|$$
$$\leq |\gamma_{S,j} - \gamma_{T,j}| \left(\alpha_j\mathcal{D}_S(\widehat{Y} = i \mid Y = j) + \beta_j\mathcal{D}_T(\widehat{Y} = i \mid Y = j)\right)$$
$$+ \gamma_{S,j} \left| \mathcal{D}_S(\widehat{Y} = i \mid Y = j) - \mathcal{D}_j(\widehat{Y} = i) \right| + \gamma_{T,j} \left| \mathcal{D}_T(\widehat{Y} = i \mid Y = j) - \mathcal{D}_j(\widehat{Y} = i) \right|.$$

To proceed, let us first simplify $\mathcal{D}_S(\widehat{Y} = i \mid Y = j) - \mathcal{D}_j(\widehat{Y} = i)$. By definition of $\mathcal{D}_j(\widehat{Y} = i) = \alpha_j\mathcal{D}_S(\widehat{Y} = i \mid Y = j) + \beta_j\mathcal{D}_T(\widehat{Y} = i \mid Y = j)$, we know that:

$$\mathcal{D}_S(\widehat{Y} = i \mid Y = j) - \mathcal{D}_j(\widehat{Y} = i)$$
$$= \mathcal{D}_S(\widehat{Y} = i \mid Y = j) - \left(\alpha_j\mathcal{D}_S(\widehat{Y} = i \mid Y = j) + \beta_j\mathcal{D}_T(\widehat{Y} = i \mid Y = j)\right)$$
$$= \left(\mathcal{D}_S(\widehat{Y} = i \mid Y = j) - \alpha_j\mathcal{D}_S(\widehat{Y} = i \mid Y = j)\right) - \beta_j\mathcal{D}_T(\widehat{Y} = i \mid Y = j)$$
$$= \beta_j\left(\mathcal{D}_S(\widehat{Y} = i \mid Y = j) - \mathcal{D}_T(\widehat{Y} = i \mid Y = j)\right).$$

Similarly, for the second term $\mathcal{D}_T(\widehat{Y} = i \mid Y = j) - \mathcal{D}_j(\widehat{Y} = i)$, we can show that:

$$\mathcal{D}_T(\widehat{Y} = i \mid Y = j) - \mathcal{D}_j(\widehat{Y} = i) = \alpha_j\left(\mathcal{D}_T(\widehat{Y} = i \mid Y = j) - \mathcal{D}_S(\widehat{Y} = i \mid Y = j)\right).$$

Plugging these two identities into the above, we can continue the analysis with

$$\left| \gamma_{S,j}(\mathcal{D}_S(\widehat{Y} = i \mid Y = j) - \mathcal{D}_j(\widehat{Y} = i)) - \gamma_{T,j}(\mathcal{D}_T(\widehat{Y} = i \mid Y = j) - \mathcal{D}_j(\widehat{Y} = i)) \right|$$
$$= \left| \gamma_{S,j}\beta(\mathcal{D}_S(\widehat{Y} = i \mid Y = j) - \mathcal{D}_T(\widehat{Y} = i \mid Y = j)) - \gamma_{T,j}\alpha_j(\mathcal{D}_T(\widehat{Y} = i \mid Y = j) - \mathcal{D}_S(\widehat{Y} = i \mid Y = j)) \right|$$
$$\leq \left| \gamma_{S,j}\beta_j(\mathcal{D}_S(\widehat{Y} = i \mid Y = j) - \mathcal{D}_T(\widehat{Y} = i \mid Y = j)) \right| + \left| \gamma_{T,j}\alpha_j(\mathcal{D}_T(\widehat{Y} = i \mid Y = j) - \mathcal{D}_S(\widehat{Y} = i \mid Y = j)) \right|$$
$$\leq \gamma_{S,j}\beta_j\Delta_{\mathrm{CE}}(\widehat{Y}) + \gamma_{T,j}\alpha_j\Delta_{\mathrm{CE}}(\widehat{Y}).$$

The first inequality holds by the triangle inequality and the second by the definition of the conditional error gap. Combining all the inequalities above completes the proof. ∎

We are now ready to prove the theorem:

**Theorem 3.1.** (Error Decomposition Theorem) For any classifier $\widehat{Y} = (h \circ g)(X)$,

$$|\varepsilon_S(h \circ g) - \varepsilon_T(h \circ g)| \leq \|\mathcal{D}_S^Y - \mathcal{D}_T^Y\|_1 \cdot \mathrm{BER}_{\mathcal{D}_S}(\widehat{Y} \parallel Y) + 2(k-1)\Delta_{\mathrm{CE}}(\widehat{Y}),$$

where $\|\mathcal{D}_S^Y - \mathcal{D}_T^Y\|_1 := \sum_{i=1}^{k} |\mathcal{D}_S(Y = i) - \mathcal{D}_T(Y = i)|$ is the $L_1$ distance between $\mathcal{D}_S^Y$ and $\mathcal{D}_T^Y$.

*Proof of Theorem 3.1.* First, by the law of total probability, it is easy to verify that following identity holds for $a \in \{S, T\}$:

$$\mathcal{D}_a(\widehat{Y} \neq Y) = \sum_{i \neq j} \mathcal{D}_a(\widehat{Y} = i, Y = j) = \sum_{i \neq j} \gamma_{a,j} \mathcal{D}_a(\widehat{Y} = i \mid Y = j).$$

Using this identity, to bound the error gap, we have:

$$
\begin{aligned}
& |\mathcal{D}_S(Y \neq \widehat{Y}) - \mathcal{D}_T(Y \neq \widehat{Y})| \\
&= \Big| \sum_{i \neq j} \gamma_{S,j} \mathcal{D}_S(\widehat{Y} = i \mid Y = j) - \sum_{i \neq j} \gamma_{T,j} \mathcal{D}_T(\widehat{Y} = i \mid Y = j) \Big| \\
&\leq \sum_{i \neq j} |\gamma_{S,j} \mathcal{D}_S(\widehat{Y} = i \mid Y = j) - \gamma_{T,j} \mathcal{D}_T(\widehat{Y} = i \mid Y = j)|.
\end{aligned}
$$

Invoking Lemma A.2 to bound the above terms, and since $\forall j \in [k], \gamma_{S,j}, \gamma_{T,j} \in [0,1], \alpha_j + \beta_j = 1$, we get:

$$
\begin{aligned}
& |\mathcal{D}_S(Y \neq \widehat{Y}) - \mathcal{D}_T(Y \neq \widehat{Y})| \\
&\leq \sum_{i \neq j} |\gamma_{S,j} \mathcal{D}_S(\widehat{Y} = i \mid Y = j) - \gamma_{T,j} \mathcal{D}_T(\widehat{Y} = i \mid Y = j)| \\
&\leq \sum_{i \neq j} |\gamma_{S,j} - \gamma_{T,j}| \cdot \Big( \alpha_j \mathcal{D}_S(\widehat{Y} = i \mid Y = j) + \beta_j \mathcal{D}_T(\widehat{Y} = i \mid Y = j) \Big) + \gamma_{S,j}\beta_j \Delta_{\mathrm{CE}}(\widehat{Y}) + \gamma_{T,j}\alpha_j \Delta_{\mathrm{CE}}(\widehat{Y}) \\
&\leq \sum_{i \neq j} |\gamma_{S,j} - \gamma_{T,j}| \cdot \Big( \alpha_j \mathcal{D}_S(\widehat{Y} = i \mid Y = j) + \beta_j \mathcal{D}_T(\widehat{Y} = i \mid Y = j) \Big) + \gamma_{S,j}\Delta_{\mathrm{CE}}(\widehat{Y}) + \gamma_{T,j}\Delta_{\mathrm{CE}}(\widehat{Y}) \\
&= \sum_{i \neq j} |\gamma_{S,j} - \gamma_{T,j}| \cdot \Big( \alpha_j \mathcal{D}_S(\widehat{Y} = i \mid Y = j) + \beta_j \mathcal{D}_T(\widehat{Y} = i \mid Y = j) \Big) + \sum_{i=1}^{k}\sum_{j \neq i} \gamma_{S,j}\Delta_{\mathrm{CE}}(\widehat{Y}) + \gamma_{T,j}\Delta_{\mathrm{CE}}(\widehat{Y}) \\
&= \sum_{i \neq j} |\gamma_{S,j} - \gamma_{T,j}| \cdot \Big( \alpha_j \mathcal{D}_S(\widehat{Y} = i \mid Y = j) + \beta_j \mathcal{D}_T(\widehat{Y} = i \mid Y = j) \Big) + 2(k-1)\Delta_{\mathrm{CE}}(\widehat{Y}).
\end{aligned}
$$

Note that the above holds $\forall \alpha_j, \beta_j \geq 0$ such that $\alpha_j + \beta_j = 1$. By choosing $\alpha_j = 1, \forall j \in [k]$ and $\beta_j = 0, \forall j \in [k]$, we have:

$$
\begin{aligned}
&= \sum_{i \neq j} |\gamma_{S,j} - \gamma_{T,j}| \cdot \mathcal{D}_S(\widehat{Y} = i \mid Y = j) + 2(k-1)\Delta_{\mathrm{CE}}(\widehat{Y}) \\
&= \sum_{j=1}^{k} |\gamma_{S,j} - \gamma_{T,j}| \cdot \Big( \sum_{i=1, i \neq j}^{k} \mathcal{D}_S(\widehat{Y} = i \mid Y = j) \Big) + 2(k-1)\Delta_{\mathrm{CE}}(\widehat{Y}) \\
&= \sum_{j=1}^{k} |\gamma_{S,j} - \gamma_{T,j}| \cdot \mathcal{D}_S(\widehat{Y} \neq Y \mid Y = j) + 2(k-1)\Delta_{\mathrm{CE}}(\widehat{Y}) \\
&\leq \|\mathcal{D}_S^Y - \mathcal{D}_T^Y\|_1 \cdot \mathrm{BER}_{\mathcal{D}_S}(\widehat{Y} \parallel Y) + 2(k-1)\Delta_{\mathrm{CE}}(\widehat{Y}),
\end{aligned}
$$

where the last line is due to Holder's inequality, completing the proof. ∎

### A.5   Proof of Theorem 3.2

**Theorem 3.2.** If $Z = g(X)$ satisfies *GLS*, then for any $h : \mathcal{Z} \to \mathcal{Y}$ and letting $\widehat{Y} = h(Z)$ be the predictor, we have $\varepsilon_S(\widehat{Y}) + \varepsilon_T(\widehat{Y}) \leq 2\mathrm{BER}_{\mathcal{D}_S}(\widehat{Y} \parallel Y)$.

*Proof.* First, by the law of total probability, we have:

$$\varepsilon_S(\widehat{Y}) + \varepsilon_T(\widehat{Y}) = \mathcal{D}_S(Y \neq \widehat{Y}) + \mathcal{D}_T(Y \neq \widehat{Y})$$

$$= \sum_{j=1}^{k}\sum_{i\neq j}\mathcal{D}_S(\widehat{Y} = i|Y = j)\mathcal{D}_S(Y = j) + \mathcal{D}_T(\widehat{Y} = i|Y = j)\mathcal{D}_T(Y = j).$$

Now, since $\widehat{Y} = (h \circ g)(X) = h(Z)$, $\widehat{Y}$ is a function of $Z$. Given the generalized label shift assumption, this guarantees that:

$$\forall y, y' \in \mathcal{Y}, \quad \mathcal{D}_S(\widehat{Y} = y' \mid Y = y) = \mathcal{D}_T(\widehat{Y} = y' \mid Y = y).$$

Thus:

$$\varepsilon_S(\widehat{Y}) + \varepsilon_T(\widehat{Y}) = \sum_{j=1}^{k}\sum_{i\neq j}\mathcal{D}_S(\widehat{Y} = i|Y = j)(\mathcal{D}_S(Y = j) + \mathcal{D}_T(Y = j))$$

$$= \sum_{j\in[k]} \mathcal{D}_S(\widehat{Y} \neq Y \mid Y = j) \cdot (\mathcal{D}_S(Y = j) + \mathcal{D}_T(Y = j))$$

$$\leq \max_{j\in[k]} \mathcal{D}_S(\widehat{Y} \neq Y \mid Y = j) \cdot \sum_{j\in[k]} \mathcal{D}_S(Y = j) + \mathcal{D}_T(Y = j)$$

$$= 2\text{BER}_{\mathcal{D}_S}(\widehat{Y} \parallel Y). \qquad \blacksquare$$

### A.6 Proof of Lemma 3.1

**Lemma 3.1.** (Necessary condition for *GLS*) If $Z = g(X)$ satisfies *GLS*, then $\mathcal{D}_T(\widetilde{Z}) = \sum_{y\in\mathcal{Y}} \mathbf{w}_y \cdot \mathcal{D}_S(\widetilde{Z}, Y = y) =: \mathcal{D}_S^{\mathbf{w}}(\widetilde{Z})$ where $\widetilde{Z}$ verifies either $\widetilde{Z} = Z$ or $\widetilde{Z} = \widehat{Y} \otimes Z$.

*Proof.* From *GLS*, we know that $\mathcal{D}_S(Z \mid Y = y) = \mathcal{D}_T(Z \mid Y = y)$. Applying any function $\tilde{h}$ to $Z$ will maintain that equality (in particular $\tilde{h}(Z) = \tilde{Y} \otimes Z$). Using that fact and Eq. (4) on the second line gives:

$$\mathcal{D}_T(\tilde{Z}) = \sum_{y\in\mathcal{Y}} \mathcal{D}_T(Y = y) \cdot \mathcal{D}_T(\tilde{Z} \mid Y = y)$$

$$= \sum_{y\in\mathcal{Y}} \mathbf{w}_y \cdot \mathcal{D}_S(Y = y) \cdot \mathcal{D}_S(\tilde{Z} \mid Y = y)$$

$$= \sum_{y\in\mathcal{Y}} \mathbf{w}_y \cdot \mathcal{D}_S(\tilde{Z}, Y = y). \qquad \blacksquare$$

### A.7 Proof of Theorem 3.3

**Theorem 3.3.** (Clustering structure implies sufficiency) Let $Z = g(X)$ such that $\mathcal{D}_T(Z) = \mathcal{D}_S^{\mathbf{w}}(Z)$. Assume $\mathcal{D}_T(Y = y) > 0, \forall y \in \mathcal{Y}$. If there exists a partition of $\mathcal{Z} = \cup_{y\in\mathcal{Y}}\mathcal{Z}_y$ such that $\forall y \in \mathcal{Y}$, $\mathcal{D}_S(Z \in \mathcal{Z}_y \mid Y = y) = \mathcal{D}_T(Z \in \mathcal{Z}_y \mid Y = y) = 1$, then $Z = g(X)$ satisfies *GLS*.

*Proof.* Follow the condition that $\mathcal{D}_T(Z) = \mathcal{D}_S^{\mathbf{w}}(Z)$, by definition of $\mathcal{D}_S^{\mathbf{w}}(Z)$, we have:

$$\mathcal{D}_T(Z) = \sum_{y\in\mathcal{Y}} \frac{\mathcal{D}_T(Y = y)}{\mathcal{D}_S(Y = y)}\mathcal{D}_S(Z, Y = y)$$

$$\Longleftrightarrow \mathcal{D}_T(Z) = \sum_{y\in\mathcal{Y}} \mathcal{D}_T(Y = y)\mathcal{D}_S(Z \mid Y = y)$$

$$\Longleftrightarrow \sum_{y\in\mathcal{Y}} \mathcal{D}_T(Y = y)\mathcal{D}_T(Z \mid Y = y) = \sum_{y\in\mathcal{Y}} \mathcal{D}_T(Y = y)\mathcal{D}_S(Z \mid Y = y).$$

Note that the above equation holds for all measurable subsets of $\mathcal{Z}$. Now by the assumption that $\mathcal{Z} = \cup_{y \in \mathcal{Y}} \mathcal{Z}_y$ is a partition of $\mathcal{Z}$, consider $\mathcal{Z}_{y'}$:

$$\sum_{y \in \mathcal{Y}} \mathcal{D}_T(Y = y) \mathcal{D}_T(Z \in \mathcal{Z}_{y'} \mid Y = y) = \sum_{y \in \mathcal{Y}} \mathcal{D}_T(Y = y) \mathcal{D}_S(Z \in \mathcal{Z}_{y'} \mid Y = y).$$

Due to the assumption $\mathcal{D}_S(Z \in \mathcal{Z}_y \mid Y = y) = \mathcal{D}_T(Z \in \mathcal{Z}_y \mid Y = y) = 1$, we know that $\forall y' \neq y$, $\mathcal{D}_T(Z \in \mathcal{Z}_{y'} \mid Y = y) = \mathcal{D}_S(Z \in \mathcal{Z}_{y'} \mid Y = y) = 0$. This shows that both the supports of $\mathcal{D}_S(Z \mid Y = y)$ and $\mathcal{D}_T(Z \mid Y = y)$ are contained in $\mathcal{Z}_y$. Now consider an arbitrary measurable set $E \subseteq \mathcal{Z}_y$, since $\cup_{y \in \mathcal{Y}} \mathcal{Z}_y$ is a partition of $\mathcal{Z}$, we know that

$$\mathcal{D}_S(Z \in E \mid Y = y') = \mathcal{D}_T(Z \in E \mid Y = y') = 0, \quad \forall y' \neq y.$$

Plug $Z \in E$ into the following identity:

$$\sum_{y \in \mathcal{Y}} \mathcal{D}_T(Y = y) \mathcal{D}_T(Z \in E \mid Y = y) = \sum_{y \in \mathcal{Y}} \mathcal{D}_T(Y = y) \mathcal{D}_S(Z \in E \mid Y = y)$$
$$\implies \mathcal{D}_T(Y = y) \mathcal{D}_T(Z \in E \mid Y = y) = \mathcal{D}_T(Y = y) \mathcal{D}_S(Z \in E \mid Y = y)$$
$$\implies \mathcal{D}_T(Z \in E \mid Y = y) = \mathcal{D}_S(Z \in E \mid Y = y),$$

where the last line holds because $\mathcal{D}_T(Y = y) \neq 0$. Realize that the choice of $E$ is arbitrary, this shows that $\mathcal{D}_S(Z \mid Y = y) = \mathcal{D}_T(Z \mid Y = y)$, which completes the proof. ∎

### A.8 Sufficient Conditions for *GLS*

**Theorem 3.4.** Let $\widehat{Y} = h(Z)$, $\gamma := \min_{y \in \mathcal{Y}} \mathcal{D}_T(Y = y)$ and $\mathbf{w}_M := \max_{y \in \mathcal{Y}} \mathbf{w}_y$. For $\widetilde{Z} = Z$ or $\widetilde{Z} = \widehat{Y} \otimes Z$, we have:

$$\max_{y \in \mathcal{Y}} d_{\mathrm{TV}}(\mathcal{D}_S(Z \mid Y = y), \mathcal{D}_T(Z \mid Y = y)) \leq \frac{\mathbf{w}_M \varepsilon_S(\widehat{Y}) + \varepsilon_T(\widehat{Y}) + \sqrt{8 D_{\mathrm{JS}}(\mathcal{D}_S^{\mathbf{w}}(\widetilde{Z}) \| \mathcal{D}_T(\widetilde{Z}))}}{\gamma}.$$

*Proof.* To prove the above upper bound, let us first fix a $y \in \mathcal{Y}$ and fix a classifier $\widehat{Y} = h(Z)$ for some $h : \mathcal{Z} \to \mathcal{Y}$. Now consider any measurable subset $E \subseteq \mathcal{Z}$, we would like to upper bound the following quantity:

$$|\mathcal{D}_S(Z \in E \mid Y = y) - \mathcal{D}_T(Z \in E \mid Y = y)|$$
$$= \frac{1}{\mathcal{D}_T(Y = y)} \cdot |\mathcal{D}_S(Z \in E, Y = y)\mathbf{w}_y - \mathcal{D}_T(Z \in E, Y = y)|$$
$$\leq \frac{1}{\gamma} \cdot |\mathcal{D}_S(Z \in E, Y = y)\mathbf{w}_y - \mathcal{D}_T(Z \in E, Y = y)|.$$

Hence it suffices if we can upper bound $|\mathcal{D}_S(Z \in E, Y = y)\mathbf{w}_y - \mathcal{D}_T(Z \in E, Y = y)|$. To do so, consider the following decomposition:

$$
\begin{aligned}
|\mathcal{D}_T(Z \in E, Y = y) - \mathcal{D}_S(Z \in E, Y = y)\mathbf{w}_y| = & |\mathcal{D}_T(Z \in E, Y = y) - \mathcal{D}_T(Z \in E, \widehat{Y} = y) \\
& + \mathcal{D}_T(Z \in E, \widehat{Y} = y) - \mathcal{D}_S^{\mathbf{w}}(Z \in E, \widehat{Y} = y) \\
& + \mathcal{D}_S^{\mathbf{w}}(Z \in E, \widehat{Y} = y) - \mathcal{D}_S(Z \in E, Y = y)\mathbf{w}_y| \\
\leq & |\mathcal{D}_T(Z \in E, Y = y) - \mathcal{D}_T(Z \in E, \widehat{Y} = y)| \\
& + |\mathcal{D}_T(Z \in E, \widehat{Y} = y) - \mathcal{D}_S^{\mathbf{w}}(Z \in E, \widehat{Y} = y)| \\
& + |\mathcal{D}_S^{\mathbf{w}}(Z \in E, \widehat{Y} = y) - \mathcal{D}_S(Z \in E, Y = y)\mathbf{w}_y|.
\end{aligned}
$$

We bound the above three terms in turn. First, consider $|\mathcal{D}_T(Z \in E, Y = y) - \mathcal{D}_T(Z \in E, \widehat{Y} = y)|$:

$$|\mathcal{D}_T(Z \in E, Y = y) - \mathcal{D}_T(Z \in E, \widehat{Y} = y)|$$
$$= |\sum_{y'} \mathcal{D}_T(Z \in E, Y = y, \widehat{Y} = y') - \sum_{y'} \mathcal{D}_T(Z \in E, \widehat{Y} = y, Y = y')|$$
$$\leq \sum_{y' \neq y} |\mathcal{D}_T(Z \in E, Y = y, \widehat{Y} = y') - \mathcal{D}_T(Z \in E, \widehat{Y} = y, Y = y')|$$
$$\leq \sum_{y' \neq y} \mathcal{D}_T(Z \in E, Y = y, \widehat{Y} = y') + \mathcal{D}_T(Z \in E, \widehat{Y} = y, Y = y')$$
$$\leq \sum_{y' \neq y} \mathcal{D}_T(Y = y, \widehat{Y} = y') + \mathcal{D}_T(\widehat{Y} = y, Y = y')$$
$$\leq \mathcal{D}_T(Y \neq \widehat{Y})$$
$$= \varepsilon_T(\widehat{Y}),$$

where the last inequality is due to the fact that the definition of error rate corresponds to the sum of all the off-diagonal elements in the confusion matrix while the sum here only corresponds to the sum of all the elements in two slices. Similarly, we can bound the third term as follows:

$$|\mathcal{D}_S^{\mathbf{w}}(Z \in E, \widehat{Y} = y) - \mathcal{D}_S(Z \in E, Y = y)\mathbf{w}_y|$$
$$= |\sum_{y'} \mathcal{D}_S(Z \in E, \widehat{Y} = y, Y = y')\mathbf{w}_{y'} - \sum_{y'} \mathcal{D}_S(Z \in E, \widehat{Y} = y', Y = y)\mathbf{w}_y|$$
$$\leq |\sum_{y' \neq y} \mathcal{D}_S(Z \in E, \widehat{Y} = y, Y = y')\mathbf{w}_{y'} - \mathcal{D}_S(Z \in E, \widehat{Y} = y', Y = y)\mathbf{w}_y|$$
$$\leq \mathbf{w}_M \sum_{y' \neq y} \mathcal{D}_S(Z \in E, \widehat{Y} = y, Y = y') + \mathcal{D}_S(Z \in E, \widehat{Y} = y', Y = y)$$
$$\leq \mathbf{w}_M \mathcal{D}_S(Z \in E, \widehat{Y} \neq Y)$$
$$\leq \mathbf{w}_M \varepsilon_S(\widehat{Y}).$$

Now we bound the last term. Recall the definition of total variation, we have:

$$|\mathcal{D}_T(Z \in E, \widehat{Y} = y) - \mathcal{D}_S^{\mathbf{w}}(Z \in E, \widehat{Y} = y)|$$
$$= |\mathcal{D}_T(Z \in E \wedge Z \in \widehat{Y}^{-1}(y)) - \mathcal{D}_S^{\mathbf{w}}(Z \in E \wedge Z \in \widehat{Y}^{-1}(y))|$$
$$\leq \sup_{E' \text{ is measurable}} |\mathcal{D}_T(Z \in E') - \mathcal{D}_S^{\mathbf{w}}(Z \in E')|$$
$$= d_{\text{TV}}(\mathcal{D}_T(Z), \mathcal{D}_S^{\mathbf{w}}(Z)).$$

Combining the above three parts yields

$$|\mathcal{D}_S(Z \in E \mid Y = y) - \mathcal{D}_T(Z \in E \mid Y = y)| \leq \frac{1}{\gamma} \cdot \left( \mathbf{w}_M \varepsilon_S(\widehat{Y}) + \varepsilon_T(\widehat{Y}) + d_{\text{TV}}(\mathcal{D}_S^{\mathbf{w}}(Z), \mathcal{D}_T(Z)) \right).$$

Now realizing that the choice of $y \in \mathcal{Y}$ and the measurable subset $E$ on the LHS is arbitrary, this leads to

$$\max_{y \in \mathcal{Y}} \sup_E |\mathcal{D}_S(Z \in E \mid Y = y) - \mathcal{D}_T(Z \in E \mid Y = y)|$$
$$\leq \frac{1}{\gamma} \cdot \left( \mathbf{w}_M \varepsilon_S(\widehat{Y}) + \varepsilon_T(\widehat{Y}) + d_{\text{TV}}(\mathcal{D}_S^{\mathbf{w}}(Z), \mathcal{D}_T(Z)) \right).$$

From Briët and Harremoës [10], we have:

$$d_{\text{TV}}(\mathcal{D}_S^{\mathbf{w}}(Z), \mathcal{D}_T(Z)) \leq \sqrt{8 D_{\text{JS}}(\mathcal{D}_S^{\mathbf{w}}(Z) \mid\mid \mathcal{D}_T(Z))}$$

(the total variation and Jensen-Shannon distance are equivalent), which gives the results for $\tilde{Z} = Z$. Finally, noticing that $z \to h(z) \otimes z$ is a bijection ($h(z)$ sums to 1), we have:

$$D_{\text{JS}}(\mathcal{D}_S^{\mathbf{w}}(Z) \mid\mid \mathcal{D}_T(Z)) = D_{\text{JS}}(\mathcal{D}_S^{\mathbf{w}}(\widehat{Y} \otimes Z) \mid\mid \mathcal{D}_T(\widehat{Y} \otimes Z)),$$

which completes the proof. ∎

Furthermore, since the above upper bound holds for any classifier $\widehat{Y} = h(Z)$, we even have:

$$\max_{y \in \mathcal{Y}} d_{\mathrm{TV}}(\mathcal{D}_S(Z \in E \mid Y = y), \mathcal{D}_T(Z \in E \mid Y = y))$$

$$\leq \frac{1}{\gamma} \cdot \inf_{\widehat{Y}} \left( \mathbf{w}_M \varepsilon_S(\widehat{Y}) + \varepsilon_T(\widehat{Y}) + d_{\mathrm{TV}}(\mathcal{D}_S^{\mathbf{w}}(Z), \mathcal{D}_T(Z)) \right).$$

### A.9 Proof of Lemma 3.2

**Lemma 3.2.** If $GLS$ is verified, and if the confusion matrix $\mathbf{C}$ is invertible, then $\mathbf{w} = \mathbf{C}^{-1}\boldsymbol{\mu}$.

*Proof.* Given (2), and with the joint hypothesis $\widehat{Y} = h(Z)$ over both source and target domains, it is straightforward to see that the induced conditional distributions over predicted labels match between the source and target domains, i.e.:

$$\mathcal{D}_S(\widehat{Y} = h(Z) \mid Y = y) = \mathcal{D}_T(\widehat{Y} = h(Z) \mid Y = y), \ \forall y \in \mathcal{Y}. \tag{14}$$

This allows us to compute $\boldsymbol{\mu}_y, \ \forall y \in \mathcal{Y}$ as

$$\mathcal{D}_T(\widehat{Y} = y) = \sum_{y' \in \mathcal{Y}} \mathcal{D}_T(\widehat{Y} = y \mid Y = y') \cdot \mathcal{D}_T(Y = y')$$

$$= \sum_{y' \in \mathcal{Y}} \mathcal{D}_S(\widehat{Y} = y \mid Y = y') \cdot \mathcal{D}_T(Y = y')$$

$$= \sum_{y' \in \mathcal{Y}} \mathcal{D}_S(\widehat{Y} = y, Y = y') \cdot \frac{\mathcal{D}_T(Y = y')}{\mathcal{D}_S(Y = y')}$$

$$= \sum_{y' \in \mathcal{Y}} \mathbf{C}_{y,y'} \cdot \mathbf{w}_{y'}.$$

where we used (14) for the second line. We thus have $\boldsymbol{\mu} = \mathbf{C}\mathbf{w}$ which concludes the proof. ∎

### A.10 $\mathcal{F}$-IPM for Distributional Alignment

In Table 4, we list different instances of IPM with different choices of the function class $\mathcal{F}$ in the above definition, including the total variation distance, Wasserstein-1 distance and the Maximum mean discrepancy [27].

Table 4: List of IPMs with different $\mathcal{F}$. $\|\cdot\|_{\mathrm{Lip}}$ denotes the Lipschitz seminorm and $\mathcal{H}$ is a reproducing kernel Hilbert space (RKHS).

| $\mathcal{F}$ | $d_{\mathcal{F}}$ |
| --- | --- |
| $\{f : \|f\|_\infty \leq 1\}$ | Total Variation |
| $\{f : \|f\|_{\mathrm{Lip}} \leq 1\}$ | Wasserstein-1 distance |
| $\{f : \|f\|_{\mathcal{H}} \leq 1\}$ | Maximum mean discrepancy |

## B  Experimentation Details

### B.1  Computational Complexity

Our algorithms imply negligible time and memory overhead compared to their base versions. They are, in practice, indistinguishable from the underlying baseline:

- Weight estimation requires storing the confusion matrix $C$ and the predictions $\mu$. This has a memory cost of $O(k^2)$, small compared to the size of a neural network that performs well on k classes.

- The extra computational cost comes from solving the quadratic program 5, which only depends on the number of classes $k$ and is solved once per epoch (not per gradient step). For Office-Home, it is a $65 \times 65$ QP, solved $\approx 100$ times. Its runtime is negligible compared to tens of thousands of gradient steps.

## B.2 Description of the domain adaptation tasks

**Digits** We follow a widely used evaluation protocol [29, 41]. For the digits datasets MNIST (M, LeCun and Cortes [32]) and USPS (U, Dheeru and Karra [19]), we consider the DA tasks: M $\rightarrow$ U and U $\rightarrow$ M. Performance is evaluated on the 10,000/2,007 examples of the MNIST/USPS test sets.

**Visda [55]** is a sim-to-real domain adaptation task. The synthetic domain contains 2D rendering of 3D models captured at different angles and lighting conditions. The real domain is made of natural images. Overall, the training, validation and test domains contain 152,397, 55,388 and 5,534 images, from 12 different classes.

**Office-31** [49] is one of the most popular dataset for domain adaptation . It contains 4,652 images from 31 classes. The samples come from three domains: Amazon (A), DSLR (D) and Webcam (W), which generate six possible transfer tasks, A $\rightarrow$ D, A $\rightarrow$ W, D $\rightarrow$ A, D $\rightarrow$ W, W $\rightarrow$ A and W $\rightarrow$ D, which we all evaluate.

**Office-Home** [54] is a more complex dataset than Office-31. It consists of 15,500 images from 65 classes depicting objects in office and home environments. The images form four different domains: Artistic (A), Clipart (C), Product (P), and Real-World images (R). We evaluate the 12 possible domain adaptation tasks.

## B.3 Full results on the domain adaptation tasks

Tables 5, 6, 7, 8, 9 and 10 show the detailed results of all the algorithms on each task of the domains described above. The performance we report is the best test accuracy obtained during training over a fixed number of epochs. We used that value for fairness with respect to the baselines (as shown in Figure 2 Left, the performance of DANN decreases as training progresses, due to the inappropriate matching of representations showcased in Theorem 2.1).

The subscript denotes the fraction of seeds for which our variant outperforms the base algorithm. More precisely, by outperform, we mean that for a given seed (which fixes the network initialization as well as the data being fed to the model) the variant has a larger accuracy on the test set than its base version. Doing so allows to assess specifically the effect of the algorithm, all else kept constant.

Table 5: Results on the Digits tasks. M and U stand for MNIST and USPS, the prefix $s$ denotes the experiment where the source domain is subsampled to increase $D_{\mathrm{JS}}(\mathcal{D}_S^Y, \mathcal{D}_T^Y)$.

| METHOD | M $\rightarrow$ U | U $\rightarrow$ M | AVG. | sM $\rightarrow$ U | sU $\rightarrow$ M | AVG. |
|---|---|---|---|---|---|---|
| NO AD. | 79.04 | 75.30 | 77.17 | 76.02 | 75.32 | 75.67 |
| DANN | 90.65 | 95.66 | 93.15 | 79.03 | 87.46 | 83.24 |
| IWDAN | $\mathbf{93.28}_{100\%}$ | $\mathbf{96.52}_{100\%}$ | $\mathbf{94.90}_{100\%}$ | $\mathbf{91.77}_{100\%}$ | $\mathbf{93.32}_{100\%}$ | $\mathbf{92.54}_{100\%}$ |
| IWDAN-O | $93.73_{100\%}$ | $96.81_{100\%}$ | $95.27_{100\%}$ | $92.50_{100\%}$ | $96.42_{100\%}$ | $94.46_{100\%}$ |
| CDAN | 94.16 | 97.29 | 95.72 | 84.91 | 91.55 | 88.23 |
| IWCDAN | $\mathbf{94.36}_{60\%}$ | $\mathbf{97.45}_{100\%}$ | $\mathbf{95.90}_{80\%}$ | $\mathbf{93.42}_{100\%}$ | $\mathbf{93.03}_{100\%}$ | $\mathbf{93.22}_{100\%}$ |
| IWCDAN-O | $94.34_{80\%}$ | $97.35_{100\%}$ | $95.85_{90\%}$ | $93.37_{100\%}$ | $96.26_{100\%}$ | $94.81_{100\%}$ |

## B.4 Jensen-Shannon divergence of the original and subsampled domain adaptation datasets

Tables 11, 12 and 13 show $D_{\mathrm{JS}}(\mathcal{D}_S(Z)||\mathcal{D}_T(Z))$ for our four datasets and their subsampled versions, rows correspond to the source domain, and columns to the target one. We recall that subsampling simply consists in taking 30% of the first half of the classes in the source domain (which explains why $D_{\mathrm{JS}}(\mathcal{D}_S(Z)||\mathcal{D}_T(Z))$ is not symmetric for the subsampled datasets).

Table 6: Results on the Visda domain. The prefix $s$ denotes the experiment where the source domain is subsampled to increase $D_{\mathrm{JS}}(\mathcal{D}_S^Y, \mathcal{D}_T^Y)$.

| METHOD | VISDA | $\|$ | $s$VISDA |
|---|---|---|---|
| NO AD. | 48.39 | $\|$ | 49.02 |
| DANN | 61.88 | $\|$ | 52.85 |
| IWDAN | $\mathbf{63.52}_{100\%}$ | $\|$ | $\mathbf{60.18}_{100\%}$ |
| IWDAN-O | $64.19_{100\%}$ | $\|$ | $62.10_{100\%}$ |
| CDAN | 65.60 | $\|$ | 60.19 |
| IWCDAN | $\mathbf{66.49}_{60\%}$ | $\|$ | $\mathbf{65.83}_{100\%}$ |
| IWCDAN-O | $68.15_{100\%}$ | $\|$ | $66.85_{100\%}$ |
| JAN | $56.98_{100\%}$ | $\|$ | $50.64_{100\%}$ |
| IWJAN | $\mathbf{57.56}_{100\%}$ | $\|$ | $\mathbf{57.12}_{100\%}$ |
| IWJAN-O | $61.48_{100\%}$ | $\|$ | $61.30_{100\%}$ |

Table 7: Results on the Office dataset.

| METHOD | A $\rightarrow$ D | A $\rightarrow$ W | D $\rightarrow$ A | D $\rightarrow$ W | W $\rightarrow$ A | W $\rightarrow$ D | AVG. |
|---|---|---|---|---|---|---|---|
| NO DA | 79.60 | 73.18 | 59.33 | 96.30 | 58.75 | 99.68 | 77.81 |
| DANN | 84.06 | 85.41 | 64.67 | 96.08 | 66.77 | 99.44 | 82.74 |
| IWDAN | $\mathbf{84.30}_{60\%}$ | $\mathbf{86.42}_{100\%}$ | $\mathbf{68.38}_{100\%}$ | $\mathbf{97.13}_{100\%}$ | $\mathbf{67.16}_{60\%}$ | $\mathbf{100.0}_{100\%}$ | $\mathbf{83.90}_{87\%}$ |
| IWDAN-O | $87.23_{100\%}$ | $88.88_{100\%}$ | $69.92_{100\%}$ | $98.09_{100\%}$ | $67.96_{80\%}$ | $99.92_{100\%}$ | $85.33_{97\%}$ |
| CDAN | $\mathbf{89.56}$ | 93.01 | 71.25 | 99.24 | 70.32 | 100.0 | 87.23 |
| IWCDAN | $88.91_{60\%}$ | $\mathbf{93.23}_{60\%}$ | $\mathbf{71.90}_{80\%}$ | $\mathbf{99.30}_{80\%}$ | $\mathbf{70.43}_{60\%}$ | $\mathbf{100.0}_{100\%}$ | $\mathbf{87.30}_{73\%}$ |
| IWCDAN-O | $90.08_{60\%}$ | $94.52_{100\%}$ | $73.11_{100\%}$ | $99.30_{80\%}$ | $71.83_{100\%}$ | $100.0_{100\%}$ | $88.14_{90\%}$ |
| JAN | 85.94 | $\mathbf{85.66}$ | $\mathbf{70.50}$ | 97.48 | $\mathbf{71.5}$ | 99.72 | 85.13 |
| IWJAN | $\mathbf{87.68}_{100\%}$ | $84.86_{0\%}$ | $70.36_{60\%}$ | $\mathbf{98.98}_{100\%}$ | $70.06_{0\%}$ | $\mathbf{100.0}_{100\%}$ | $\mathbf{85.32}_{60\%}$ |
| IWJAN-O | $89.68_{100\%}$ | $89.18_{100\%}$ | $71.96_{100\%}$ | $99.02_{100\%}$ | $73.0_{100\%}$ | $100.0_{100\%}$ | $87.14_{100\%}$ |

## B.5 Losses

### B.5.1 DANN

For batches of data $(x_S^i, y_S^i)$ and $(x_T^i)$ of size $s$, the DANN losses are:

$$\mathcal{L}_{DA}(x_S^i, y_S^i, x_T^i; \theta, \psi) = -\frac{1}{s}\sum_{i=1}^{s} \log(d_\psi(g_\theta(x_S^i))) + \log(1 - d_\psi(g_\theta(x_T^i))), \quad (15)$$

$$\mathcal{L}_{C}(x_S^i, y_S^i; \theta, \phi) = -\frac{1}{s}\sum_{i=1}^{s} \log(h_\phi(g_\theta(x_S^i)_{y_S^i})). \quad (16)$$

### B.5.2 CDAN

Similarly, the CDAN losses are:

$$\mathcal{L}_{DA}(x_S^i, y_S^i, x_T^i; \theta, \psi) = -\frac{1}{s}\sum_{i=1}^{s} \log(d_\psi(h_\phi(g_\theta(x_S^i)) \otimes g_\theta(x_S^i))) \quad (17)$$

$$+ \log(1 - d_\psi(h_\phi(g_\theta(x_T^i)) \otimes g_\theta(x_T^i))), \quad (18)$$

$$\mathcal{L}_{C}(x_S^i, y_S^i; \theta, \phi) = -\frac{1}{s}\sum_{i=1}^{s} \log(h_\phi(g_\theta(x_S^i)_{y_S^i})), \quad (19)$$

where $h_\phi(g_\theta(x_S^i)) \otimes g_\theta(x_S^i) := (h_1(g(x_S^i))g(x_S^i), \ldots, h_k(g(x_S^i))g(x_S^i))$ and $h_1(g(x_S^i))$ is the $i$-th element of vector $h(g(x_S^i))$.

Table 8: Results on the Subsampled Office dataset.

| METHOD | sA → D | sA → W | sD → A | sD → W | sW → A | sW → D | Avg. |
|---|---|---|---|---|---|---|---|
| No DA | 75.82 | 70.69 | 56.82 | 95.32 | 58.35 | 97.31 | 75.72 |
| DANN | 75.46 | 77.66 | 56.58 | 93.76 | 57.51 | 96.02 | 76.17 |
| IWDAN | $\mathbf{81.61}_{100\%}$ | $\mathbf{88.43}_{100\%}$ | $\mathbf{65.00}_{100\%}$ | $\mathbf{96.98}_{100\%}$ | $\mathbf{64.86}_{100\%}$ | $\mathbf{98.72}_{100\%}$ | $\mathbf{82.60}_{100\%}$ |
| IWDAN-O | $84.94_{100\%}$ | $91.17_{100\%}$ | $68.44_{100\%}$ | $97.74_{100\%}$ | $64.57_{100\%}$ | $99.60_{100\%}$ | $84.41_{100\%}$ |
| CDAN | 82.45 | 84.60 | 62.54 | 96.83 | 65.01 | 98.31 | 81.62 |
| IWCDAN | $\mathbf{86.59}_{100\%}$ | $\mathbf{87.30}_{100\%}$ | $\mathbf{66.45}_{100\%}$ | $\mathbf{97.69}_{100\%}$ | $\mathbf{66.34}_{100\%}$ | $\mathbf{98.92}_{100\%}$ | $\mathbf{83.88}_{100\%}$ |
| IWCDAN-O | $87.39_{100\%}$ | $91.47_{100\%}$ | $69.69_{100\%}$ | $97.91_{100\%}$ | $67.50_{100\%}$ | $98.88_{100\%}$ | $85.47_{100\%}$ |
| JAN | 77.74 | 77.64 | 64.48 | 91.68 | 92.60 | 65.10 | 78.21 |
| IWJAN | $\mathbf{84.62}_{100\%}$ | $\mathbf{83.28}_{100\%}$ | $\mathbf{65.30}_{80\%}$ | $\mathbf{96.30}_{100\%}$ | $\mathbf{98.80}_{100\%}$ | $\mathbf{67.38}_{100\%}$ | $\mathbf{82.61}_{97\%}$ |
| IWJAN-O | $88.42_{100\%}$ | $89.44_{100\%}$ | $72.06_{100\%}$ | $97.26_{100\%}$ | $98.96_{100\%}$ | $71.30_{100\%}$ | $86.24_{100\%}$ |

Table 9: Results on the Office-Home dataset.

| METHOD | A → C | A → P | A → R | C → A | C → P | C → R | |
|---|---|---|---|---|---|---|---|
| No DA | 41.02 | 62.97 | 71.26 | 48.66 | 58.86 | 60.91 | |
| DANN | 46.03 | 62.23 | 70.57 | 49.06 | 63.05 | 64.14 | |
| IWDAN | $\mathbf{48.65}_{100\%}$ | $\mathbf{69.19}_{100\%}$ | $\mathbf{73.60}_{100\%}$ | $\mathbf{53.59}_{100\%}$ | $\mathbf{66.25}_{100\%}$ | $\mathbf{66.09}_{100\%}$ | |
| IWDAN-O | $50.19_{100\%}$ | $70.53_{100\%}$ | $75.44_{100\%}$ | $56.69_{100\%}$ | $67.40_{100\%}$ | $67.98_{100\%}$ | |
| CDAN | 49.00 | 69.23 | 74.55 | 54.46 | 68.23 | 68.9 | |
| IWCDAN | $\mathbf{49.81}_{100\%}$ | $\mathbf{73.41}_{100\%}$ | $\mathbf{77.56}_{100\%}$ | $\mathbf{56.5}_{100\%}$ | $69.64_{80\%}$ | $\mathbf{70.33}_{100\%}$ | |
| IWCDAN-O | $52.31_{100\%}$ | $74.54_{100\%}$ | $78.46_{100\%}$ | $60.33_{100\%}$ | $70.78_{100\%}$ | $71.47_{100\%}$ | |
| JAN | $\mathbf{41.64}$ | 67.20 | 73.12 | 51.02 | 62.52 | 64.46 | |
| IWJAN | $41.12_{0\%}$ | $\mathbf{67.56}_{80\%}$ | $\mathbf{73.14}_{60\%}$ | $\mathbf{51.70}_{100\%}$ | $\mathbf{63.42}_{100\%}$ | $\mathbf{65.22}_{100\%}$ | |
| IWJAN-O | $41.88_{80\%}$ | $68.72_{100\%}$ | $73.62_{100\%}$ | $53.04_{100\%}$ | $63.88_{100\%}$ | $66.48_{100\%}$ | |
| METHOD | P → A | P → C | P → R | R → A | R → C | R → P | Avg. |
| No DA | 47.1 | 35.94 | 68.27 | 61.79 | 44.42 | 75.5 | 56.39 |
| DANN | 48.29 | 44.06 | 72.62 | 63.81 | 53.93 | 77.64 | 59.62 |
| IWDAN | $\mathbf{52.81}_{100\%}$ | $\mathbf{46.24}_{80\%}$ | $\mathbf{73.97}_{100\%}$ | $\mathbf{64.90}_{100\%}$ | $\mathbf{54.02}_{80\%}$ | $\mathbf{77.96}_{100\%}$ | $\mathbf{62.27}_{97\%}$ |
| IWDAN-O | $59.33_{100\%}$ | $48.28_{100\%}$ | $76.37_{100\%}$ | $69.42_{100\%}$ | $56.09_{100\%}$ | $78.45_{100\%}$ | $64.68_{100\%}$ |
| CDAN | 56.77 | $\mathbf{48.8}$ | 76.83 | $\mathbf{71.27}$ | $\mathbf{55.72}$ | $\mathbf{81.27}$ | 64.59 |
| IWCDAN | $\mathbf{58.99}_{100\%}$ | $48.41_{0\%}$ | $\mathbf{77.94}_{100\%}$ | $69.48_{0\%}$ | $54.73_{0\%}$ | $81.07_{60\%}$ | $\mathbf{65.66}_{70\%}$ |
| IWCDAN-O | $62.60_{100\%}$ | $50.73_{100\%}$ | $78.88_{100\%}$ | $72.44_{100\%}$ | $57.79_{100\%}$ | $81.31_{80\%}$ | $67.64_{98\%}$ |
| JAN | 54.5 | 40.36 | $\mathbf{73.10}$ | $\mathbf{64.54}$ | $\mathbf{45.98}$ | $\mathbf{76.58}$ | 59.59 |
| IWJAN | $\mathbf{55.26}_{80\%}$ | $\mathbf{40.38}_{60\%}$ | $73.08_{80\%}$ | $64.40_{60\%}$ | $45.68_{0\%}$ | $76.36_{40\%}$ | $\mathbf{59.78}_{63\%}$ |
| IWJAN-O | $57.78_{100\%}$ | $41.32_{100\%}$ | $73.66_{100\%}$ | $65.40_{100\%}$ | $46.68_{100\%}$ | $76.36_{20\%}$ | $60.73_{92\%}$ |

CDAN is particularly well-suited for conditional alignment. As described in Section 2, the CDAN discriminator seeks to match $\mathcal{D}_S(\widehat{Y} \otimes Z)$ with $\mathcal{D}_T(\widehat{Y} \otimes Z)$. This objective is very aligned with $GLS$: let us first assume for argument's sake that $\widehat{Y}$ is a perfect classifier on both domains. For any sample $(x, y)$, $\hat{y} \otimes z$ is thus a matrix of 0s except on the $y$-th row, which contains $z$. When label distributions match, the effect of fooling the discriminator will result in representations such that the matrices $\widehat{Y} \otimes Z$ are equal on the source and target domains. In other words, the model is such that $Z \mid Y$ match: it verifies $GLS$ (see Th. 3.4 below with $\mathbf{w} = 1$). On the other hand, if the label distributions differ, fooling the discriminator actually requires mislabelling certain samples (a fact quantified in Th. 2.1).

Table 10: Results on the subsampled Office-Home dataset.

| METHOD | A → C | A → P | A → R | C → A | C → P | C → R |
|---|---|---|---|---|---|---|
| NO DA | 35.70 | 54.72 | 62.61 | 43.71 | 52.54 | 56.62 |
| DANN | 36.14 | 54.16 | 61.72 | 44.33 | 52.56 | 56.37 |
| IWDAN | $39.81_{100\%}$ | $63.01_{100\%}$ | $68.67_{100\%}$ | $47.39_{100\%}$ | $61.05_{100\%}$ | $60.44_{100\%}$ |
| IWDAN-O | $42.79_{100\%}$ | $66.22_{100\%}$ | $71.40_{100\%}$ | $53.39_{100\%}$ | $61.47_{100\%}$ | $64.97_{100\%}$ |
| CDAN | 38.90 | 56.80 | 64.77 | 48.02 | 60.07 | 61.17 |
| IWCDAN | $42.96_{100\%}$ | $65.01_{100\%}$ | $71.34_{100\%}$ | $52.89_{100\%}$ | $64.65_{100\%}$ | $66.48_{100\%}$ |
| IWCDAN-O | $45.76_{100\%}$ | $68.61_{100\%}$ | $73.18_{100\%}$ | $56.88_{100\%}$ | $66.61_{100\%}$ | $68.48_{100\%}$ |
| JAN | 34.52 | 56.86 | 64.54 | 46.18 | 56.84 | 59.06 |
| IWJAN | $36.24_{100\%}$ | $61.00_{100\%}$ | $66.34_{100\%}$ | $48.66_{100\%}$ | $59.92_{100\%}$ | $61.88_{100\%}$ |
| IWJAN-O | $37.46_{100\%}$ | $62.68_{100\%}$ | $66.88_{100\%}$ | $49.82_{100\%}$ | $60.22_{100\%}$ | $62.54_{100\%}$ |

| METHOD | P → A | P → C | P → R | R → A | R → C | R → P | AVG. |
|---|---|---|---|---|---|---|---|
| NO DA | 44.29 | 33.05 | 65.20 | 57.12 | 40.46 | 70.0 | |
| DANN | 44.58 | 37.14 | 65.21 | 56.70 | 43.16 | 69.86 | 51.83 |
| IWDAN | $50.44_{100\%}$ | $41.63_{100\%}$ | $72.46_{100\%}$ | $61.00_{100\%}$ | $49.40_{100\%}$ | $76.07_{100\%}$ | $57.61_{100\%}$ |
| IWDAN-O | $56.05_{100\%}$ | $43.39_{100\%}$ | $74.87_{100\%}$ | $66.73_{100\%}$ | $51.72_{100\%}$ | $77.46_{100\%}$ | $60.87_{100\%}$ |
| CDAN | 49.65 | 41.36 | 70.24 | 62.35 | 46.98 | 74.69 | 56.25 |
| IWCDAN | $54.87_{100\%}$ | $44.80_{100\%}$ | $75.91_{100\%}$ | $67.02_{100\%}$ | $50.45_{100\%}$ | $78.55_{100\%}$ | $61.24_{100\%}$ |
| IWCDAN-O | $59.63_{100\%}$ | $46.98_{100\%}$ | $77.54_{100\%}$ | $69.24_{100\%}$ | $53.77_{100\%}$ | $78.11_{100\%}$ | $63.73_{100\%}$ |
| JAN | 50.64 | 37.24 | 69.98 | 58.72 | 40.64 | 72.00 | 53.94 |
| IWJAN | $52.92_{100\%}$ | $37.68_{100\%}$ | $70.88_{100\%}$ | $60.32_{100\%}$ | $41.54_{100\%}$ | $73.26_{100\%}$ | $55.89_{100\%}$ |
| IWJAN-O | $56.54_{100\%}$ | $39.66_{100\%}$ | $71.78_{100\%}$ | $62.36_{100\%}$ | $44.56_{100\%}$ | $73.76_{100\%}$ | $57.36_{100\%}$ |

Table 11: Jensen-Shannon divergence between the label distributions of the Digits and Visda tasks.

(A) FULL DATASET

| | MNIST | USPS | REAL |
|---|---|---|---|
| MNIST | 0 | 6.64e−3 | - |
| USPS | 6.64e−3 | 0 | - |
| SYNTH. | - | - | 2.61e−2 |

(B) SUBSAMPLED

| | MNIST | USPS | REAL |
|---|---|---|---|
| MNIST | 0 | 6.52e−2 | - |
| USPS | 2.75e−2 | 0 | - |
| SYNTH. | - | - | 6.81e−2 |

### B.5.3 JAN

The JAN losses [40] are :

$$\mathcal{L}_{DA}(x_S^i, y_S^i, x_T^i; \theta, \psi) = -\frac{1}{s^2}\sum_{i,j=1}^{s} k(x_S^i, x_S^j) - \frac{1}{s^2}\sum_{i,j=1}^{s} k(x_T^i, x_T^j) + \frac{2}{s^2}\sum_{i,j=1}^{s} k(x_S^i, x_T^j) \quad (20)$$

$$\mathcal{L}_C(x_S^i, y_S^i; \theta, \phi) = -\frac{1}{s}\sum_{i=1}^{s} \log(h_\phi(g_\theta(x_S^i)_{y_S^i})), \quad (21)$$

where $k$ corresponds to the kernel of the RKHS $\mathcal{H}$ used to measure the discrepancy between distributions. Exactly as in Long et al. [40], it is the product of kernels on various layers of the network $k(x_S^i, x_S^j) = \prod_{l \in \mathcal{L}} k^l(x_S^i, x_S^j)$. Each individual kernel $k^l$ is computed as the dot-product between

Table 12: Jensen-Shannon divergence between the label distributions of the Office-31 tasks.

(A) FULL DATASET

|  | AMAZON | DSLR | WEBCAM |
|---|---|---|---|
| AMAZON | 0 | 1.76e−2 | 9.52e−3 |
| DSLR | 1.76e−2 | 0 | 2.11e−2 |
| WEBCAM | 9.52e−3 | 2.11e−2 | 0 |

(B) SUBSAMPLED

|  | AMAZON | DSLR | WEBCAM |
|---|---|---|---|
| AMAZON | 0 | 6.25e−2 | 4.61e−2 |
| DSLR | 5.44e−2 | 0 | 5.67e−2 |
| WEBCAM | 5.15e−2 | 7.05e−2 | 0 |

Table 13: Jensen-Shannon divergence between the label distributions of the Office-Home tasks.

(A) FULL DATASET

|  | ART | CLIPART | PRODUCT | REAL WORLD |
|---|---|---|---|---|
| ART | 0 | 3.85e−2 | 4.49e−2 | 2.40e−2 |
| CLIPART | 3.85e−2 | 0 | 2.33e−2 | 2.14e−2 |
| PRODUCT | 4.49e−2 | 2.33e−2 | 0 | 1.61e−2 |
| REAL WORLD | 2.40e−2 | 2.14e−2 | 1.61e−2 | 0 |

(B) SUBSAMPLED

|  | ART | CLIPART | PRODUCT | REAL WORLD |
|---|---|---|---|---|
| ART | 0 | 8.41e−2 | 8.86e−2 | 6.69e−2 |
| CLIPART | 7.07e−2 | 0 | 5.86e−2 | 5.68e−2 |
| PRODUCT | 7.85e−2 | 6.24e−2 | 0 | 5.33e−2 |
| REAL WORLD | 6.09e−2 | 6.52e−2 | 5.77e−2 | 0 |

two transformations of the representation: $k^l(x_S^i, x_S^j) = \langle d_\psi^l(g_\theta^l(x_S^i)), d_\psi^l(g_\theta^l(x_S^j)) \rangle$ (in this case, $d_\psi^l$ outputs vectors in a high-dimensional space). See Section B.7 for more details.

The IWJAN losses are:

$$\mathcal{L}_{DA}^{\mathbf{w}}(x_S^i, y_S^i, x_T^i; \theta, \psi) = -\frac{1}{s^2} \sum_{i,j=1}^{s} \mathbf{w}_{y_S^i} \mathbf{w}_{y_S^j} k(x_S^i, x_S^j) - \frac{1}{s^2} \sum_{i,j=1}^{s} k(x_T^i, x_T^j) + \frac{2}{s^2} \sum_{i,j=1}^{s} \mathbf{w}_{y_S^i} k(x_S^i, x_T^j)$$

(22)

$$\mathcal{L}_C^{\mathbf{w}}(x_S^i, y_S^i; \theta, \phi) = -\frac{1}{s} \sum_{i=1}^{s} \frac{\mathbf{w}_{y_S^i}}{k\mathcal{D}_S(Y = y)} \log(h_\phi(g_\theta(x_S^i))_{y_S^i}).$$

(23)

## B.6 Generation of domain adaptation tasks with varying $D_{\mathbf{JS}}(\mathcal{D}_S(Z) \parallel \mathcal{D}_T(Z))$

We consider the MNIST → USPS task and generate a set $\mathcal{V}$ of 50 vectors in $[0.1, 1]^{10}$. Each vector corresponds to the fraction of each class to be trained on, either in the source or the target domain (to assess the impact of both). The left bound is chosen as 0.1 to ensure that classes all contain some samples.

This methodology creates 100 domain adaptation tasks, 50 for *subsampled*-MNIST → USPS and 50 for MNIST → *subsampled*-USPS, with Jensen-Shannon divergences varying from 6.1e−3 to

9.53e$-2$[2]. They are then used to evaluate our algorithms, see Section 4 and Figures 1 and 3. They show the performance of the 6 algorithms we consider. We see the sharp decrease in performance of the base versions DANN and CDAN. Comparatively, our importance-weighted algorithms maintain good performance even for large divergences between the marginal label distributions.

## B.7  Implementation details

All the values reported below are the default ones in the implementations of DANN, CDAN and JAN released with the respective papers (see links to the github repos in the footnotes). We did not perform any search on them, assuming they had already been optimized by the authors of those papers. To ensure a fair comparison and showcase the simplicity of our approach, we simply plugged the weight estimation on top of those baselines and used their original hyperparameters.

For MNIST and USPS, the architecture is akin to LeNet [31], with two convolutional layers, ReLU and MaxPooling, followed by two fully connected layers. The representation is also taken as the last hidden layer, and has 500 neurons. The optimizer for those tasks is SGD with a learning rate of 0.02, annealed by 0.5 every five training epochs for M $\rightarrow$ U and 6 for U $\rightarrow$ M. The weight decay is also 5e$-4$ and the momentum 0.9.

For the Office and Visda experiments with IWDAN and IWCDAN, we train a ResNet-50, optimized using SGD with momentum. The weight decay is also 5e$-4$ and the momentum 0.9. The learning rate is 3e$-4$ for the Office-31 tasks A $\rightarrow$ D and D $\rightarrow$ W, 1e$-3$ otherwise (default learning rates from the CDAN implementation[3]).

For the IWJAN experiments, we use the default implementation of Xlearn codebase[4] and simply add the weigths estimation and reweighted objectives to it, as described in Section B.5. Parameters, configuration and networks remain the same.

Finally, for the Office experiments, we update the importance weights $\mathbf{w}$ every 15 passes on the dataset (in order to improve their estimation on small datasets). On Digits and Visda, the importance weights are updated every pass on the source dataset. Here too, fine-tuning that value might lead to a better estimation of $\mathbf{w}$ and help bridge the gap with the oracle versions of the algorithms.

We use the cvxopt package[5] to solve the quadratic programm 5.

We trained our models on single-GPU machines (P40s and P100s). The runtime of our algorithms is undistinguishable from the the runtime of their base versions.

## B.8  Weight Estimation

We estimate importance weights using Lemma 3.2, which relies on the *GLS* assumption. However, there is no guarantee that *GLS* is verified at any point during training, so the exact dynamics of $\mathbf{w}$ are unclear. Below we discuss those dynamics and provide some intuition about them.

In Fig. 4b, we plot the Euclidian distance between the moving average of weights estimated using the equation $\mathbf{w} = \mathbf{C}^{-1}\boldsymbol{\mu}$ and the true weights (note that this can be done for any algorithm). As can be seen in the figure, the distance between the estimated and true weights is highly correlated with the performance of the algorithm (see Fig.4). In particular, we see that the estimations for IWDAN is more accurate than for DANN. The estimation for DANN exhibits an interesting shape, improving at first, and then getting worse. At the same time, the estimation for IWDAN improves monotonously. The weights for IWDAN-O get very close to the true weights which is in line with our theoretical results: IWDAN-O gets close to zero error on the target error, Th. 3.4 thus guarantees that *GLS* is verified, which in turns implies that the weight estimation is accurate (Lemma 3.2). Finally, without domain adaptation, the estimation is very poor. The following two lemmas shed some light on the phenomena observed for DANN and IWDAN:

**Lemma 3.3.** If the source error $\varepsilon_S(h \circ g)$ is zero and the source and target marginals verify $D_{\mathrm{JS}}\big(\mathcal{D}_S^{\tilde{\mathbf{w}}}(Z), \mathcal{D}_T(Z)\big) = 0$, then the estimated weight vector $\mathbf{w}$ is equal to $\tilde{\mathbf{w}}$.

(a) Performance of DANN, IWDAN and IWDAN-O.  (b) Performance of CDAN, CDAN and IWCDAN.

Figure 3: Performance in % of our algorithms and their base versions. The $x$-axis represents $D_{\text{JS}}(\mathcal{D}_S^Y, \mathcal{D}_T^Y)$, the Jensen-Shannon distance between label distributions. Lines represent linear fits to the data. For both sets of algorithms, the larger the jsd, the larger the improvement.

*Proof.* If $\varepsilon_S(h \circ g) = 0$, then the confusion matrix $\mathbf{C}$ is diagonal and its $y$-th line is $\mathcal{D}_S(Y = y)$. Additionally, if $D_{\text{JS}}(\mathcal{D}_S^{\tilde{\mathbf{w}}}(Z), \mathcal{D}_T(Z)) = 0$, then from a straightforward extension of Eq. 12, we have $D_{\text{JS}}(\mathcal{D}_S^{\tilde{\mathbf{w}}}(\hat{Y}), \mathcal{D}_T(\hat{Y})) = 0$. In other words, the distribution of predictions on the source and target domains match, *i.e.* $\boldsymbol{\mu}_y = \mathcal{D}_T(\hat{Y} = y) = \sum_{y'} \tilde{\mathbf{w}}_{y'} \mathcal{D}_S(\hat{Y} = y, Y = y') = \tilde{\mathbf{w}}_y \mathcal{D}_S(Y = y), \forall y$ (where the last equality comes from $\varepsilon_S(h \circ g) = 0$). Finally, we get that $\mathbf{w} = \mathbf{C}^{-1}\boldsymbol{\mu} = \tilde{\mathbf{w}}$. ∎

In particular, applying this lemma to DANN (*i.e.* with $\tilde{\mathbf{w}}_{y'} = \mathbf{1}$) suggests that at convergence, the estimated weights should tend to $\mathbf{1}$. Empirically, Fig. 4b shows that as the marginals get matched, the estimation for DANN does get closer to $\mathbf{1}$ ($\mathbf{1}$ corresponds to a distance of 2.16)[6]. We now attempt to provide some intuition on the behavior of IWDAN, with the following lemma:

**Lemma B.1.** If $\varepsilon_S(h \circ g) = 0$ and if for a given $y$:

$$\min(\tilde{\mathbf{w}}_y \mathcal{D}_S(Y = y), \mathcal{D}_T(Y = y)) \leq \boldsymbol{\mu}_y \leq \max(\tilde{\mathbf{w}}_y \mathcal{D}_S(Y = y), \mathcal{D}_T(Y = y)), \quad (24)$$

then, letting $\mathbf{w} = \mathbf{C}^{-1}\boldsymbol{\mu}$ be the estimated weight:

$$|\mathbf{w}_y - \mathbf{w}_y^*| \leq |\tilde{\mathbf{w}}_y - \mathbf{w}_y^*|.$$

Applying this lemma to $\tilde{\mathbf{w}}_y = \mathbf{w}_t$, and assuming that (24) holds for all the classes $y$ (we discuss what the assumption implies below), we get that:

$$\|\mathbf{w}_{t+1} - \mathbf{w}_y^*\| \leq \|\mathbf{w}_t - \mathbf{w}_y^*\|, \quad (25)$$

or in other words, the estimation improves monotonously. Combining this with Lemma B.1 suggests an explanation for the shape of the IWDAN estimated weights on Fig. 4b: the monotonous improvement of the estimation is counter-balanced by the matching of weighted marginals which, when reached, makes $\mathbf{w}_t$ constant (Lemma 3.3 applied to $\tilde{\mathbf{w}} = \mathbf{w}_t$). However, we wish to emphasize that the exact dynamics of $\mathbf{w}$ are complex, and we do not claim understand them fully. In all likelihood, they are the by-product of regularity in the data, properties of deep neural networks and their interaction with stochastic gradient descent. Additionally, the dynamics are also inherently linked to the success of domain adaptation, which to this day remains an open problem.

As a matter of fact, assumption (24) itself relates to successful domain adaptation. Setting aside $\tilde{\mathbf{w}}$, which simply corresponds to a class reweighting of the source domain, (24) states that predictions on the target domain fall between a successful prediction (corresponding to $\mathcal{D}_T(Y = y)$) and the prediction of a model with matched marginals (corresponding to $\mathcal{D}_S(Y = y)$). In other words, we

assume that the model is naturally in between successful domain adaptation and successful marginal matching. Empirically, we observed that it holds true for most classes (with $\tilde{\mathbf{w}} = \tilde{\mathbf{w}}_t$ for IWDAN and with $\tilde{\mathbf{w}} = \mathbf{1}$ for DANN), but not all early in training[7].

To conclude this section, we prove Lemma B.1.

*Proof.* From $\varepsilon_S(h \circ g) = 0$, we know that $\mathbf{C}$ is diagonal and that its $y$-th line is $\mathcal{D}_S(Y = y)$. This gives us: $\mathbf{w}_y = (\mathbf{C}^{-1}\boldsymbol{\mu})_y = \frac{\mu_y}{\mathcal{D}_S(Y=y)}$. Hence:

$$\min(\tilde{\mathbf{w}}_y \mathcal{D}_S(Y = y), \mathcal{D}_T(Y = y)) \leq \mu_y \leq \max(\tilde{\mathbf{w}}_y \mathcal{D}_S(Y = y), \mathcal{D}_T(Y = y))$$

$$\iff \frac{\min(\tilde{\mathbf{w}}_y \mathcal{D}_S(Y = y), \mathcal{D}_T(Y = y))}{\mathcal{D}_S(Y = y)} \leq \frac{\mu_y}{\mathcal{D}_S(Y = y)} \leq \frac{\max(\tilde{\mathbf{w}}_y \mathcal{D}_S(Y = y), \mathcal{D}_T(Y = y))}{\mathcal{D}_S(Y = y)}$$

$$\iff \min(\tilde{\mathbf{w}}_y, \mathbf{w}_y^*) \leq \mathbf{w}_y \leq \max(\tilde{\mathbf{w}}_y, \mathbf{w}_y^*)$$

$$\iff \min(\tilde{\mathbf{w}}_y, \mathbf{w}_y^*) - \mathbf{w}_y^* \leq \mathbf{w}_y - \mathbf{w}_y^* \leq \max(\tilde{\mathbf{w}}_y, \mathbf{w}_y^*) - \mathbf{w}_y^*$$

$$\iff |\mathbf{w}_y - \mathbf{w}_y^*| \leq |\tilde{\mathbf{w}}_y - \mathbf{w}_y^*|,$$

which conludes the proof. ∎

(a) Transfer accuracy during training.   (b) Distance to true weights during training.

Figure 4: *Left* Accuracy of various algorithms during training. *Right* Euclidian distance between the weights estimated using Lemma 3.2 and the true weights. Those plots correspond to averages over 5 seeds.

## B.9   Per-class predictions and estimated weights

In this section, we display the per-class predictions of various algorithms on the sU → M task. In Table 16, we see that without domain adaptation, performance on classes is rather random, the digit 9 for instance is very poorly predicted and confused with 4 and 8.

Table 17 shows an interesting pattern for DANN. In line with the limitations described by Theorem 2.1, the model performs very poorly on the subsampled classes (we recall that the subsampling is done in the source domain): the neural network tries to match the unweighted marginals. To do so, it projects representations of classes that are over-represented in the target domain (digits 0 to 4) on representations of the under-represented classes (digits 5 to 9). In doing so, it heavily degrades its performance on those classes (it is worth noting that digit 0 has an importance weight close to 1 which probably explains why DANN still performs well on it, see Table 14).

As far as IWDAN is concerned, Table 18 shows that the model perfoms rather well on all classes, at the exception of the digit 7 confused with 9. IWDAN-O is shown in Table 19 and as expected outperforms the other algorithms on all classes.

Finally, Table 14 shows the estimated weights of all the algorithms, at the training epoch displayed in Tables 16, 17, 18 and 19. We see a rather strong correlation between errors on the estimated weight for a given class, and errors in the predictions for that class (see for instance digit 3 for DANN or digit 7 for IWDAN).

Table 14: Estimated weights and their euclidian distance to the true weights, taken at the training epoch for the confusion matrices in Tables 16, 17, 18 and 19. The first row contains the true weights. The last column gives the euclidian distance from the true weights.

| | CLASS | | | | | | | | | | |
| | 0 | 1 | 2 | 3 | 4 | 5 | 6 | 7 | 8 | 9 | DISTANCE |
|---|---|---|---|---|---|---|---|---|---|---|---|
| TRUE | 1.19 | 1.61 | 1.96 | 2.24 | 2.16 | 0.70 | 0.64 | 0.70 | 0.78 | 0.66 | 0 |
| DANN | 1.06 | 1.15 | 1.66 | 1.33 | 1.95 | 0.86 | 0.72 | 0.70 | 1.02 | 0.92 | 1.15 |
| IWDAN | 1.19 | 1.61 | 1.92 | 1.96 | 2.31 | 0.70 | 0.63 | 0.55 | 0.78 | 0.78 | 0.38 |
| IWDAN-O | 1.19 | 1.60 | 2.01 | 2.14 | 2.1 | 0.73 | 0.64 | 0.65 | 0.78 | 0.66 | 0.12 |
| NO DA | 1.14 | 1.4 | 2.42 | 1.49 | 4.21 | 0.94 | 0.38 | 0.82 | 0.62 | 0.29 | 2.31 |

Table 15: Ablation study on the Digits tasks, with weights learnt during training.

| Method | Digits | sDigits | Method | Digits | sDigits |
|---|---|---|---|---|---|
| DANN | 93.15 | 83.24 | CDAN | 95.72 | 88.23 |
| DANN + $\mathcal{L}_C^{\mathbf{w}}$ | 93.18 | 84.20 | CDAN + $\mathcal{L}_C^{\mathbf{w}}$ | 95.30 | 91.14 |
| DANN + $\mathcal{L}_{DA}^{\mathbf{w}}$ | **94.35** | **92.48** | CDAN + $\mathcal{L}_{DA}^{\mathbf{w}}$ | 95.42 | 92.35 |
| IWDAN | **94.90** | **92.54** | IWCDAN | **95.90** | **93.22** |

Table 16: Per-class predictions without domain adaptation on the sU → M task. Average accuracy: 74.49%. The table $M$ below verifies $M_{ij} = \mathcal{D}_T(\hat{Y} = j | Y = i)$.

| | | | | | | | | | |
|---|---|---|---|---|---|---|---|---|---|
| 92.89 | 0.13 | 3.24 | 0.00 | 2.20 | 0.01 | 0.45 | 0.88 | 0.18 | 0.02 |
| 0.00 | 72.54 | 12.38 | 0.00 | 3.40 | 0.37 | 7.54 | 1.50 | 2.28 | 0.00 |
| 0.31 | 0.23 | 93.28 | 0.09 | 0.72 | 0.03 | 0.34 | 4.78 | 0.17 | 0.05 |
| 0.06 | 0.77 | 4.81 | 68.53 | 1.50 | 19.91 | 0.02 | 2.48 | 1.61 | 0.31 |
| 0.02 | 0.62 | 0.28 | 0.00 | 97.19 | 0.51 | 0.04 | 0.17 | 0.79 | 0.37 |
| 0.75 | 3.03 | 0.69 | 1.01 | 1.20 | 88.96 | 0.39 | 0.31 | 2.69 | 0.96 |
| 0.73 | 1.98 | 0.42 | 0.03 | 23.86 | 2.74 | 69.08 | 0.29 | 0.12 | 0.75 |
| 1.02 | 2.01 | 4.16 | 0.13 | 9.32 | 6.36 | 0.01 | 73.48 | 1.01 | 2.50 |
| 6.01 | 8.27 | 2.55 | 1.35 | 1.62 | 3.62 | 4.98 | 6.96 | 64.40 | 0.24 |
| 1.49 | 3.35 | 0.55 | 1.28 | 38.30 | 15.36 | 0.05 | 20.68 | 1.34 | 17.60 |

Table 17: Per-class predictions for DANN on the sU → M task. Average accuracy: 86.71%. The table $M$ below verifies $M_{ij} = \mathcal{D}_T(\hat{Y} = j|Y = i)$. The first 5 classes are under-represented in the source domain compared to the target domain. On those (except 0), DANN does not perform as well as on the over-represented classes (the last 5). In line with Th. 2.1, matching the representation distributions on source and target forced the classifier to confuse the digits "1", "3" and "4" in the target domain with "8", "5" and "9".

| | | | | | | | | | |
|---|---|---|---|---|---|---|---|---|---|
| 95.79 | 0.01 | 0.08 | 0.01 | 0.12 | 0.38 | 2.34 | 0.36 | 0.36 | 0.57 |
| 0.14 | 70.77 | 0.80 | 0.01 | 1.03 | 1.29 | 9.46 | 0.06 | 16.39 | 0.06 |
| 1.61 | 0.14 | 89.82 | 0.20 | 0.42 | 0.48 | 0.83 | 3.73 | 1.37 | 1.39 |
| 0.46 | 0.08 | 1.10 | 63.33 | 0.04 | 26.28 | 0.02 | 1.78 | 3.76 | 3.14 |
| 0.11 | 0.13 | 0.05 | 0.00 | 78.17 | 0.85 | 0.16 | 0.15 | 1.97 | 18.41 |
| 0.19 | 0.04 | 0.02 | 0.04 | 0.01 | 91.30 | 0.25 | 0.27 | 5.97 | 1.91 |
| 0.62 | 0.12 | 0.01 | 0.00 | 1.98 | 4.61 | 91.73 | 0.05 | 0.36 | 0.51 |
| 0.14 | 0.23 | 1.39 | 0.13 | 0.10 | 0.32 | 0.02 | 94.10 | 1.46 | 2.10 |
| 0.69 | 0.13 | 0.11 | 0.05 | 0.21 | 2.12 | 0.50 | 0.36 | 95.19 | 0.66 |
| 0.36 | 0.31 | 0.03 | 0.08 | 0.46 | 3.67 | 0.01 | 1.64 | 1.03 | 92.40 |

Table 18: Per-class predictions for IWDAN on the sU → M task. Average accuracy: 94.38%. The table $M$ below verifies $M_{ij} = \mathcal{D}_T(\hat{Y} = j|Y = i)$.

| | | | | | | | | | |
|---|---|---|---|---|---|---|---|---|---|
| 97.33 | 0.06 | 0.23 | 0.01 | 0.20 | 0.43 | 1.29 | 0.19 | 0.18 | 0.09 |
| 0.00 | 97.71 | 0.41 | 0.05 | 0.56 | 0.69 | 0.14 | 0.03 | 0.34 | 0.07 |
| 0.70 | 0.16 | 96.32 | 0.08 | 0.34 | 0.23 | 0.23 | 1.49 | 0.43 | 0.01 |
| 0.23 | 0.01 | 0.97 | 87.67 | 0.01 | 9.25 | 0.02 | 0.63 | 0.87 | 0.35 |
| 0.11 | 0.25 | 0.05 | 0.00 | 96.93 | 0.16 | 0.22 | 0.02 | 0.40 | 1.85 |
| 0.15 | 0.11 | 0.01 | 0.16 | 0.05 | 95.81 | 0.69 | 0.11 | 2.82 | 0.08 |
| 0.26 | 0.25 | 0.00 | 0.00 | 2.07 | 1.49 | 95.84 | 0.01 | 0.07 | 0.00 |
| 0.16 | 0.42 | 2.12 | 0.91 | 1.15 | 0.60 | 0.03 | 82.07 | 1.35 | 11.19 |
| 0.44 | 0.50 | 0.36 | 0.18 | 0.43 | 0.90 | 0.91 | 0.15 | 95.74 | 0.40 |
| 0.34 | 0.42 | 0.06 | 0.30 | 1.67 | 2.46 | 0.11 | 0.31 | 0.85 | 93.50 |

Table 19: Per-class predictions for IWDAN-O on the sU $\rightarrow$ M task. Average accuracy: 96.8%. The table $M$ below verifies $M_{ij} = \mathcal{D}_T(\hat{Y} = j | Y = i)$.

| | | | | | | | | | |
|---|---|---|---|---|---|---|---|---|---|
| 98.04 | 0.01 | 0.20 | 0.00 | 0.27 | 0.03 | 1.17 | 0.11 | 0.15 | 0.02 |
| 0.00 | 98.35 | 0.33 | 0.15 | 0.17 | 0.27 | 0.19 | 0.05 | 0.47 | 0.02 |
| 0.22 | 0.04 | 97.48 | 0.07 | 0.29 | 0.08 | 0.52 | 1.09 | 0.19 | 0.02 |
| 0.10 | 0.00 | 0.66 | 95.72 | 0.01 | 2.32 | 0.00 | 0.35 | 0.56 | 0.27 |
| 0.01 | 0.25 | 0.05 | 0.00 | 96.80 | 0.03 | 0.18 | 0.01 | 0.60 | 2.06 |
| 0.23 | 0.11 | 0.00 | 0.72 | 0.00 | 96.09 | 0.68 | 0.13 | 2.01 | 0.03 |
| 0.27 | 0.31 | 0.00 | 0.00 | 2.07 | 0.63 | 96.54 | 0.00 | 0.17 | 0.01 |
| 0.26 | 0.45 | 2.13 | 0.29 | 0.90 | 0.32 | 0.01 | 92.66 | 1.06 | 1.92 |
| 0.55 | 0.22 | 0.30 | 0.06 | 0.18 | 0.22 | 0.41 | 0.33 | 97.11 | 0.62 |
| 0.46 | 0.37 | 0.16 | 0.86 | 0.82 | 1.45 | 0.01 | 0.77 | 0.98 | 94.13 |