[Reviews · NeurIPS 2020]

Review 1

Summary and Contributions: This paper proposes an approach based on importance weighting to perform unsupervised domain adaptation under a setting where there is a mismatch between the label distributions from source and target domains. Motivated by previous work that showed that, in case label shift exists, learning domain-invariant representations is actually harmful to the target domain performance in binary classification tasks, the authors extended these results for multi-class classification problems and devised generalization bounds on the target performance which depend on a more general label shift condition. The authors further build on the devised generalization guarantees to propose a domain adaptation strategy that depends on reweighting the source distribution. This strategy can be included in previous domain adaptation methods and the empirical analysis showed that it overall provides improvements in the target performance.

Strengths: - The work is well-motivated and proposes a theoretically grounded approach to address unsupervised domain adaptation problems under label shift. - Most of the contributions found in this work are motivated by either previous or new theoretical results. - The results presented in Section 3 boils down to a practical approach presented in Section 4, which seems easy to implement and to incorporate to previously proposed domain adaptation strategies.

Weaknesses: The main weaknesses of this work lie in the empirical validation of the proposed approach. - After comparing the results presented in Tables 2 and 3, it seems that the gain obtained with the proposed importance weighting is mostly coming from applying it to the adversarial component of the loss. More importantly, in cases such as DANN + L^w_DA (Table 2) vs IWDANN (Table 3) on both Digits and sDigits tasks, using importance weighting on both components of the losses is hurting the performance (95.31 vs 94.90). In this case, DANN + L^w_DA is even slightly better than the IWDAN-O. It is not clear why this is the case and this aspect should be discussed in the paper. - It is not clear from either the main paper or the supplementary material how the empirical evaluation was performed (see “Correctness” for details).

Correctness: - It is not possible to evaluate whether the empirical methodology is correct. There are important details missing in the manuscript such as how hyperparameter search was performed for all the domain adaptation strategies and the corresponding importance weighted (IW) versions. Important questions to be answered: 1- How were the hyperparameters reported in Appendix B.7 found? 2- Were the same values used for the baselines and the IW versions? My main concern here is whether the comparison with the considered baseline is fair, i.e. were the baselines also tuned using the same strategy/budget as the IW versions? - It is also not clear how the performance of each method is assessed. Is early stopping being performed or each model is trained by a fixed number epochs and the reported performance corresponds to the last epoch? In case early stopping is being used, what is the adopted stopping criterion? Please add this information to the paper, otherwise it is not possible to reproduce the results reported in this work.

Clarity: The paper is overall well written, but there are some inconsistencies in the notation that if fixed can make it easier to follow (see "Additional feedback" for details).

Relation to Prior Work: The authors did not include in the related work section prior work in the broad domain adaptation literature that also proposed to use reweighting strategies to deal with domain mismatches (a few examples are [1, 2, 3, 4]). I believe it is important to highlight the differences with respect to prior work in terms of the considered aspects such as the considered setting, the goals of reweigthing, and the strategies to estimate the weights. [1] Chen, Qingchao, et al. "Re-weighted adversarial adaptation network for unsupervised domain adaptation." Proceedings of the IEEE Conference on Computer Vision and Pattern Recognition. 2018. [2] Zhang, Jing, et al. "Importance weighted adversarial nets for partial domain adaptation." Proceedings of the IEEE Conference on Computer Vision and Pattern Recognition. 2018. [3] Matsuura, Toshihiko, Kuniaki Saito, and Tatsuya Harada. "TWINs: Two weighted inconsistency-reduced networks for partial domain adaptation." arXiv preprint arXiv:1812.07405 (2018). [4] Bouvier, Victor, et al. "Hidden Covariate Shift: A Minimal Assumption For Domain Adaptation." arXiv preprint arXiv:1907.12299 (2019).

Reproducibility: Yes

Additional Feedback: In the following, I present a list of questions and suggestions. - Line 4: "In this paper, we propose generalized label shift (GLS) to improve robustness against mismatched label distributions.". This is statement is vague. Please specify whether GLS is an assumption or a condition you aim to achieve. - Line 31: “Our contributions are the following. First, we extend the upper bound by Zhao et al. [62] to k-class classification and to conditional domain adversarial networks, a recently introduced domain adaptation algorithm”. It is not clear to me why extending the upper bound in [62] to CDAN could be seen as a contribution. - Line 47: “We focus on the general k-class classification problem” what “general” means here? - Lines 53 and 60: overloaded notation. h was first defined as a hypothesis on the input space (from X to [k]. I would use Y instead of [k] here to be consitent) and then the same h was used to denote a hypothesis on the feature space (from Z to Y). - Line 58: add respectively to indicate that labeled points are sampled from D_S and unlabeled points sampled from D_T. - Line 58: “Inspired by Ben-David et al. [7], a common approach is to learn representations invariant to the domain shift”. Please specify which kind of domain shift. - Line 67: Specify that z=g(x). - The authors introduced in line 56 a notation for the marginal distributions (D^X_S and D^Y_S in the case of the source domain), however, whenever the argument of D_S is explicit in the text, the superscript indicating the marginal is omitted (e.g. Eq. 2). Even though the argument makes it possible to infer whether the joint or marginals are being considered, I found this a bit confusing. I think it would be more clear if the notation with the superscript was adopted throughout all the equations whenever marginal distributions are considered. - Line 125: "We now provide performance guarantees for models that satisfy GLS", what are the models referred here? It is not totally clear to me what exactly needs to satisfy the GLS. Is it the representation space Z? -Line 213: "The sections above suggest" above doesn't make sense here. Replace it by "previous sections". - Section 4.2 is a bit cluttered (especially on page 7) and figures 1 and 2 are small and a little hard to read. - Axes labels on Figures 1, 2, 3, and 4 should be properly capitalized. - The Appendix is quite long it would be helpful to include a summary in the beginning. Moreover, it would also be helpful to have a sketch of proof for Theorems 3.1 and 3.4 so that it is possible to have a general idea of the main steps. Minor: Labelled (line 15) vs Labeled (line 57): pick one and be consistent throughout the text. - Line 70: missing space “B.5).Building”. - Line 139: "The second is \Delta_{CE}(Y) measures" connector missing here. -------Update after rebuttal------- I have read the rebuttal and it partially addressed my concerns. I would have appreciated if the authors had included in their response at least a short paragraph highlighting the main differences between their contribution and the closely related work pointed out in the review (the blank rows in the rebuttal document could have been used for that). I, therefore, decided to keep my score but modified the "Reproducibility" checkbox to "Yes" assuming that the authors will include in the manuscript the experimental details provided in the rebuttal.


Review 2

Summary and Contributions: (1) This paper develops a theoretical foundation for domain adaptation. Different from the common covariate shift and label shift assumptions, it proposes the generalized label shift (GLS) assumption. Sufficient and necessary conditions of GLS are discussed, and a novel error decomposition is derived. (2) Based on the theory, it proposes an algorithm that matches features using the importance weights of labels. (3) Experiments on common datasets show that the proposed methods improve over the corresponding counterparts. ======= Update after rebuttal ======= I would like to thank the authors for the feedback. For the Markov chain argument in Thm.2.1, the one-hot classifier *is* the block selection operator. Even though they can be defined differently on the algorithmic level as mentioned in the proof/rebuttal, they are not independent variables. Thus I still consider the Markov chain argument to be inaccurate.

Strengths: - Novel theoretical analytics based on GLS - Practical algorithms - Promising experiment results

Weaknesses: Some of the discussions could be more specific.

Correctness: There is a problem with the proof of Thm.2.1. The Markov chain mentioned in L565 is incorrect since \tilde{Z} also depends on \hat{Y}. In fact, CDAN uses Z to get \hat{Y}, and the \tilde{Z} is only used for the discriminator. This will also change subsequent discussions. If the oracle is using ground-truth weight (as mentioned in L234), why doesn't it have zero all the time in Fig.2 right?

Clarity: L113: Can you elaborate on why a noiseless h^* satisfies GLS? L232: Can you elaborate on how the weighting in Eq.(7) corresponds to the BER in Thm.3.1? The BER in Thm.3.1 is defined w.r.t. the source domain, which corresponds to the denominator in Eq.(7), but where does the importance weight w come from?

Relation to Prior Work: Yes.

Reproducibility: Yes

Additional Feedback:


Review 3

Summary and Contributions: 1. Generalizes the upper bound proposed by Zhao et al. [62] to multi-class classification and general feature space Z. 2. Introduces generalized label shift (GLS) in which conditional invariance is placed in representation rather than input space. 3. Provides performance guarantees for algorithms that seek to enforce GLS.

Strengths: 1. Ample theoretical frameworks that endeavors to understand domain adaptation problem. 2. Provides how to circumvent the mismatched label distributions problem by importance reweighting.

Weaknesses: 1. One of the major concerns of this work is that the assumptions throughout the paper might be too strict to be applied to practical applications. The condition for GLS and the clustering condition in Theorem 3.3 are not seemed to be possible in real domain adaptation scenarios. 2. The authors claim that the merit of the proposed Theorem 3.1 is that it does not require knowledge about unknown optimal labeling function in feature space, but \Delta_CE(Y^) needs the information about the ground truth labels of samples from its definition. Thus, the proposed Theorem 3.1 is considered to be a mere transformed form of Theorem 2 of [7] where the core notion of each term is the same. 3. The main contribution of the proposed method is the importance reweighting to circumvent the label distribution shift problem, but the performance improvement is minute which hampers the contribution of this work. In practical use, is it more important than solving mode collapsing in adversarial domain adaptation or label switching in conditional distribution matching algorithms? In particular, to satisfy GLS, the label switching problem should be averted, and this is not addressed in this work.

Correctness: I could not find any critical defect on the proofs of the presented theorems.

Clarity: The manuscript is well written and easy to follow. The supplemental material is properly provided.

Relation to Prior Work: The novelty of this work compared to related prior work is properly addressed.

Reproducibility: Yes

Additional Feedback: Post-rebuttal. Thank you for the thorough response. I decided to raise my score to 6 considering the reasonable explanations about the concerns.


Review 4

Summary and Contributions: This paper addresses the label shift issue of the domain adaptation on the perspective of the relaxed condition. The authors upper bound the target risk through the degree of the generalized label shift and the balanced error rate of the source domain. Then, the authors define importance weights and the reweighted marginal distribution and show that matching the target feature distribution with the reweighted source distribution is a proper objective of the label shift problem. Then, practical approaches with confusion matrix are introduced to empirically verify the theoretic demonstrations.

Strengths: - The definition of GSL is reasonable, and discussions on it seem necessary. The authors reasonably expand and explain the theorems and lemmas derived from the new definition. - Experimental results support the necessity to consider GSL. - The authors sufficiently provide proof and analysis in the supplemental material.

Weaknesses: - The practical implementation quit seems to rely on the previous work [1]. But the weaknesses do not outweigh the novelty of the approach. [1] Detecting and Correcting for Label Shift with Black Box Predictors, Zachary C. Lipton, Yu-Xiang Wang, Alexander J. Smola, ICML'18.

Correctness: The claims,method, and proof look reasonable and correct.

Clarity: Overall well-written. Clear and reasonable explanation with sufficient proof in the supplementary material.

Relation to Prior Work: It might be better to clearly explain the difference between the proposed practical method and [1].

Reproducibility: Yes

Additional Feedback:

[Author Response · NeurIPS 2020]

We would like to thank all the reviewers for their time and comments, which will definitely help improve the paper.
We now provide a point-by-point response to the comments.

**[R1, Ablation study from Table 2]:** The numbers reported in the table were obtained with the real weights, not the
estimated ones (so performance needs to be compared to IWDAN-O, not IWDAN). We wanted to deconfound the
effects of weight estimation effects from the improvements provided by the two losses. To confirm things, we have
run the ablation with weight estimation. On e.g. Digits, DANN+$\mathcal{L}_{DA}^w$ has a perf. of $94.35\%$, which is lower than
IWDAN ($94.90\%$). We will clarify this and add an ablation table to the paper with results using weight estimation.

**[R1, Hyperparameter selection]:** The values reported in B.7 are the default ones in the implementations of DANN,
CDAN and JAN released with the respective papers (the links to the github repos are provided in B.7). We did not
perform any search on them, assuming they had already been optimized by the authors of those papers. To ensure a
fair comparison and showcase the simplicity of our approach, we simply plugged the weight estimation on top of the
baselines with their original hyperparameters, and did not optimize those for IW.

**[R1, Performance]:** The performance we report is the best test accuracy obtained during training over a fixed number
of epochs (we will specify that number in the appendix). We used that value for fairness with respect to baselines (as
shown in Fig.2 Left, the performance of DANN decreases as training progresses, due to the inappropriate matching of
representations showcased in Th.2.1). Thank you for the various suggestions and missing references/discussion, we
will include them in the next iteration.

**[R2, Markov chain]:** We would like to clarify that the $\hat{Y}$ mentioned by the reviewer corresponds to $h(Z)$, the one-hot
classifier, in Line 565, and the $\hat{Y}$ used in Line 565 is the block selection operator defined over $\tilde{Z}$, so the Markov chain
still holds and as a result Theorem 2.1 is correct. We thank the reviewer for clarifying the CDAN algorithm, and we are
happy to update our discussion about CDAN (consequently also the presentation of Theorem 2.1) in our next iteration.

**[R2, Weight distance]:** In Fig.2 Right, the weights reported are computed using the confusion matrix and the predic-
tions on the target domain as described in Lemma 3.2. For IWDAN-O, those weights are not used, just computed, and
the non-zero distance is because at initialization GLS is not verified. As training progresses, the model becomes closer
and closer to verifying GLS, and as such, the distance to the true weights goes to 0. We will clarify this in the text.

**[R2, $h^*$ satisfying GLS]:** This is because the existence of ground-truth labeling function means that for each $X$, there
will be only one true label $Y$, hence conditioning on $Y$ essentially partitions the input space of $X$. As a result, the
conditional distribution of $h^*(X) \mid Y = y$ reduces to a point mass (a Dirac distribution) concentrated on $y$ over both
domains, trivially satisfying GLS.

**[R2, Reweighting in Eq.7]:** The $w$ appears in the numerator under the assumption of GLS, we will clarify this in our
next iteration. If GLS is not verified, as the reviewer pointed out, there would be no $w$ in the numerator.

**[R3, GLS hard to hold in practice]:** We agree with the reviewer that GLS is hard to satisfy *exactly* in practice, but the
goal of Theorem 3.1 and Theorem 3.4 is to inspire our algorithmic design and neither of these two theorems requires
GLS to hold exactly.

**[R3, Theorem 3.1]:** We respectfully disagree with this comment. As we discuss from Lines 140 to 146, our error
decomposition is completely orthogonal (hence not a mere transformation) to the existing one (Theorem 2 of [7]).
Essentially, they correspond to two different ways of decomposing a joint distribution, i.e., Ours: $\Pr(X, Y) = \Pr(Y) \cdot$
$\Pr(X \mid Y)$ versus Existing: $\Pr(X, Y) = \Pr(X) \cdot \Pr(Y \mid X)$, hence the core notions of each term are completely
different. Furthermore, our claim is also correct. The $\Delta_{\mathrm{CE}}$ is about the conditional distribution of $\Pr(Z \mid Y)$ whereas
the optimal labeling function is $\Pr(Y \mid Z)$.

**[R3, Label switching and mode collapse]:** The problem we aim to tackle in this paper is orthogonal to those two
issues. We do not think it is more important, we simply think that having models robust to mismatched label dis-
tributions is important for successful domain adaptation (and in the case of large mismatches, the improvement our
algorithms provide is rather significant - Table 3, subsampled datasets). We agree that our work is not a silver bullet
to all the problems DA encounters, but we do believe our method can augment most (if not all) algorithms designed to
improve performance in DA.

**[R5, Relation to [1]]:** Thanks for your kind comments. We will add a clearer comparison with [1] in our final version.
At a high level, both the method in [1] and ours for importance weight estimation use the core idea of moment matching
under GLS. However, the specific algorithms used to obtain these estimations are different. In particular, the method
in [1] uses the inversion of the estimated confusion matrix directly, while ours proposes to solve the QP in Eq. (4).
This difference is very important in practice: matrix inversion is notoriously unstable and the estimated weight by
matrix inversion could be infeasible, i.e., the weight could be negative. Our QP formulation explicitly get rids of both
issues hence is more numerically stable and guarantees to return feasible importance weights.

[Meta-Review · NeurIPS 2020]

This paper proposes a new approach to unsupervised domain adaptation (UDA) under label shift. The idea is a generalized label shift (GLS) assumption where conditional invariance is placed in representation rather than input space. The main contributions include 1) generalizing the information-theoretic lower bound of error to multiple classes; 2) devising generalization bounds in the target domain based on the balanced error rate and conditional error gap; 3) deriving necessary and sufficient conditions for GLS; 4) efficient importance reweighting algorithm for target/source label distributions using the integral probability metric. Overall, all reviewers including myself find the GLS framework interesting, providing an important new approach to UDA that can be flexibility embedded in existing methods. The theoretical foundation is also solid. The main concern is in the experiment, for which the author’s rebuttal has provided useful details. I recommend that the authors include the related information from the rebuttal to the final version of the paper, and further clarify 1) the Markov chain argument in Thm.2.1; 2) the connection and differences between the paper and the related work as pointed out by the reviewers.